# eQMARL: Entangled Quantum Multi-Agent Reinforcement Learning for Distributed Cooperation over Quantum Channels

**Alexander DeRieux** ⬤   **Walid Saad** ⬤

Bradley Department of Electrical and Computer Engineering, Virginia Tech, USA
`{derieux,walids}@vt.edu`

## Abstract

Collaboration is a key challenge in distributed multi-agent reinforcement learning (MARL) environments. Learning frameworks for these decentralized systems must weigh the benefits of explicit player coordination against the communication overhead and computational cost of sharing local observations and environmental data. Quantum computing has sparked a potential synergy between quantum entanglement and cooperation in multi-agent environments, which could enable more efficient distributed collaboration with minimal information sharing. This relationship is largely unexplored, however, as current state-of-the-art quantum MARL (QMARL) implementations rely on classical information sharing rather than entanglement over a quantum channel as a coordination medium. In contrast, in this paper, a novel framework dubbed entangled QMARL (eQMARL) is proposed. The proposed eQMARL is a distributed actor-critic framework that facilitates cooperation over a quantum channel and eliminates local observation sharing via a quantum entangled split critic. Introducing a quantum critic uniquely spread across the agents allows coupling of local observation encoders through entangled input qubits over a quantum channel, which requires no explicit sharing of local observations and reduces classical communication overhead. Further, agent policies are tuned through joint observation-value function estimation via joint quantum measurements, thereby reducing the centralized computational burden. Experimental results show that eQMARL with $\Psi^+$ entanglement converges to a cooperative strategy up to $17.8\%$ faster and with a higher overall score compared to split classical and fully centralized classical and quantum baselines. The results also show that eQMARL achieves this performance with a constant factor of 25-times fewer centralized parameters compared to the split classical baseline.

## 1 Introduction

Quantum reinforcement learning (QRL) is emerging as a relatively new class of quantum machine learning (QML) for decision making. Exploiting the performance and data encoding enhancements of quantum computing, QRL has many promising applications across diverse areas such as finance (Herman et al., 2022), healthcare (Flöther, 2023), and even wireless networks (Narottama et al., 2023). Its multi-agent variant, quantum multi-agent reinforcement learning (QMARL), is of specific interest because of the potential synergies between decentralized agent cooperation and quantum entanglement. Indeed, in quantum mechanics (Einstein et al., 1971), entanglement is a distinctly quantum property that intrinsically links the behavior of one particle with another regardless of their physical proximity. The use of entanglement in the broader field of QML is a recent notion. Few core works like Mitarai et al. (2018) and Du et al. (2020) use entangled layers within variational quantum circuit (VQC) designs to link the behavior of *quantum bits* (or *qubits*) within a single hybrid quantum model. Even in the recently proposed quantum split learning (QSL) framework (Yun et al., 2023a), entanglement is only used locally within each VQC branch of the quantum split neural network (QSNN) model. What has not yet been explored, however, is using entanglement to *couple* the behavior of multiple QML models. In QMARL, the use of entanglement can be further extended to the implicit coordination amongst agents during training time. Historically, in both purely classical and quantum multi-agent reinforcement learning (MARL), classical communication, shared

replay buffers, centralized global networks, and fully-observable environment assumptions have all proven to be viable methods for coordinating a group policy (Yun et al., 2022a; 2023b; 2022b; Chen, 2023; Park et al., 2023; Kölle et al., 2024). Even QSL, which is not exclusive to MARL, relies fully on classical communication between branches of the QSNN (Yun et al., 2023a). None of these approaches, however, take advantage of the available quantum channel and quantum entanglement as coupling mediums across decentralized agents or model branches, and opt instead for more classical methods of coordination. In short, entanglement is one such phenomenon of quantum mechanics that has not yet been fully explored in the context of cooperation in QMARL settings.

In contrast to prior art, we propose a novel framework dubbed *entangled QMARL (eQMARL)*. The proposed eQMARL is a distributed actor-critic framework, intersecting canonical centralized training with decentralized execution (CTDE) and fully decentralized learning, that facilitates collaboration over a quantum channel using a quantum entangled split critic. Our design uniquely allows agents to coordinate their policies by, for the first time, splitting the quantum critic architecture over a quantum channel and coupling their localized observation encoders using entangled input qubits. This uniquely allows agents to cooperate over a quantum channel, which eliminates the need for observation sharing, and further reduces classical communication overhead. Also, agent policies are tuned via joint observation-value function estimation using joint quantum measurements across all qubits in the system, which minimizes the computational burden of a central server. As will be evident from our analysis, eQMARL will be shown to converge to a cooperative strategy faster, with higher overall score on average, and with fewer centralized parameters compared to baselines. All of our source code and experiments are publicly available on GitHub[1].

## 1.1 RELATED WORKS AND THEIR LIMITATIONS

QMARL is a nascent field, with few works applying the quantum advantage to scenarios with multiple agents (Yun et al., 2022a; 2023b; 2022b; Chen, 2023; Park et al., 2023; Kölle et al., 2024). Further, the application of quantum to split learning (SL) is even newer, with Yun et al. (2023a) being the only prior work. A complete overview of prior works is provided as supplementary material in Appendix A. The resounding theme in these prior works is the use of independent agents or branches that communicate and learn through centralized classical means. No prior work, however, makes use of the quantum channel as a medium for system coupling or for multi-agent collaboration. Indeed, the quantum elements serve as drop-in replacements for classical neural network (NN) counterparts, and, importantly, the quantum channel between agents and the potential for sharing entangled qubit states go largely under-utilized. Simply put, entanglement and the quantum channel are potentially useful untapped cooperative resources intrinsic to QMARL that have largely unknown benefits.

## 1.2 CONTRIBUTIONS

The contributions of this work are summarized as follows:

- We propose a novel eQMARL framework that trains decentralized policies via a split quantum critic over a quantum channel with entangled input qubits and joint measurement.
- We propose a new QMARL algorithm for training distributed agents via optimizing a split critic *without sharing local environment observations amongst agents or a central server*.
- We show that the split nature of eQMARL reduces the computational burden of a central quantum server by distributing and tuning parameterized quantum gates across agents in the system, and requiring a small number of parameters for joint measurement.
- We empirically demonstrate that eQMARL with $\Psi^+$ entanglement exhibits a faster convergence time that can reach up to $17.8\%$ faster, and with higher overall score, compared to split classical and fully centralized classical and quantum baselines in environments with full and partial information. Further, the results also show that eQMARL achieves this level of performance and cooperation with a constant factor of $25$-times fewer centralized parameters compared to the split classical baseline.

To the best of our knowledge, this is the first application of QMARL that exploits the quantum channel and entanglement between agents to learn without sharing local observations, while also reducing the classical communication overhead and central computation burden of leading approaches.

---

[1]`https://github.com/news-vt/eqmarl`

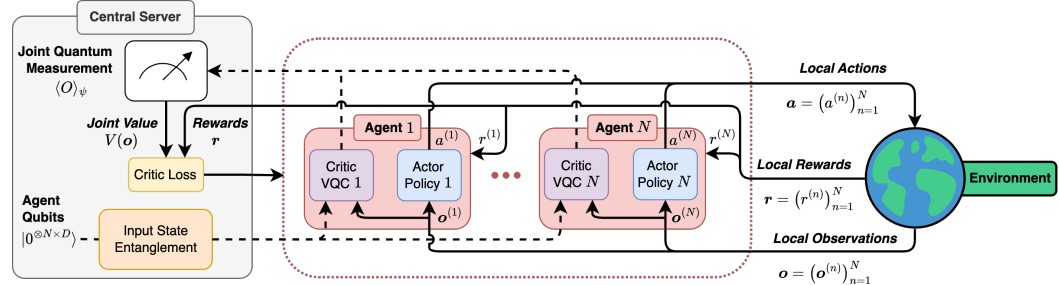

Figure 1: General design of our eQMARL framework. Dashed (solid) arrows represent quantum (classical) communication. A split quantum critic is deployed across the agents via local VQCs (purple) coupled via input entanglement (orange) at a trusted central server (gray). Joint quantum measurements across all qubits (white) estimate the joint value from the locally-encoded observations.

## 2 PROPOSED EQMARL FRAMEWORK

Our proposed eQMARL framework is a new approach for training multi-agent actor-critic architectures which lies at the intersection between CTDE and fully decentralized learning. Inspired from CTDE, we deploy decentralized agent policies which learn using a joint value function at training time. The key to our approach, however, is that we use quantum entanglement to deploy the joint value function estimator as a critic network *which is spread across the agents to operate in mostly decentralized fashion*. An overview of our framework design is shown in Fig. 1, and the design of the system architecture from a purely quantum perspective is shown in Fig. 2. From Fig. 1, the two main elements of eQMARL are a *central quantum server* and a set of $N$ *decentralized quantum agents* $\mathcal{N} = \{n\}_{n=1}^{N}$. The decentralized agents do not communicate with each other; only communication with the server is necessary during training. During execution, the agents interact with the environment independently and are fully decentralized. During training, our eQMARL framework is divided into core stages: 1) Centralized quantum input state entanglement preparation, 2) Decentralized agent environment observation encoding and variational rotations, and 3) Joint value estimation through joint quantum measurement. Fig. 2 shows how input states are prepared using custom pairwise entanglement operators, followed by agent VQCs, and then joint measurements. Physically, these operations occur at different locations, however, it is equivalent to consider these as a single quantum system, from input state preparation to final measurement. For purposes of quantum state transmission, we assume an ideal quantum channel environment with no losses.

### 2.1 JOINT INPUT ENTANGLEMENT

The first stage of eQMARL creates an entangled input state for the split quantum critic network, which couples the critic VQCs spread across the agents. To understand how this works, we first provide a brief primer on quantum entanglement. More comprehensive preliminaries are provided as supplementary material in Appendix B. Consider two independent agents $A$ and $B$, with quantum systems described by Hilbert spaces $\mathcal{H}_A$ and $\mathcal{H}_B$, and arbitrary quantum states $|\psi\rangle_A$ and $|\psi\rangle_B$. The combined Hilbert space of the two systems can be represented using the Kronecker (i.e., tensor) product $\mathcal{H}_{AB} = \mathcal{H}_A \otimes \mathcal{H}_B$. The combined quantum system is said to be *separable* if the agent states can be cleanly separated as a tensor product of the two systems, i.e., $|\psi\rangle_{AB} = |\psi\rangle_A \otimes |\psi\rangle_B$. If, however, the states cannot be represented in this form, then the combined system is said to be inseparable, or *entangled*. For example, if each agent has one qubit in the $\{|0\rangle, |1\rangle\}$ basis, then a separable system could have the state $|\psi_{\text{separable}}\rangle_{AB} = |0\rangle_A \otimes |1\rangle_A$, whereas an entangled system could have the state $|\psi_{\text{entangled}}\rangle_{AB} = (|00\rangle_{AB} + |11\rangle_{AB})/\sqrt{2}$. In eQMARL, each agent $n \in \mathcal{N}$ is assigned a set of $D$ qubits $\mathcal{Q}^{(n)} = \{q_d^{(n)}\}_{d=1}^{D}$ chosen based upon the environment state dimension and desired quantum state encoding method. The total number of system qubits is $N \times D$, and is represented by the union of agent qubit sets $\mathcal{Q} = \bigcup_{n=1}^{N} \mathcal{Q}^{(n)}$. We couple the agents by preparing an input state that pairwise entangles their qubits using a variation of Bell state entanglement (Nielsen & Chuang, 2012) such that

$$\text{ENT}_{\delta(1,d),\dots,\delta(N,d)}^{B} = \begin{cases} \left(\bigotimes_{n=2}^{N} \text{CNOT}_{\delta(1,d),\delta(n,d)}\right) H_{\delta(1,d)}, & \text{if } B = \Phi^+, \\ \left(\bigotimes_{n=2}^{N} \text{CNOT}_{\delta(1,d),\delta(n,d)}\right) H_{\delta(1,d)} X_{\delta(1,d)}, & \text{if } B = \Phi^-, \\ \left(\bigotimes_{n=2}^{N} \text{CNOT}_{\delta(1,d),\delta(n,d)}\right) H_{\delta(1,d)} \left(\bigotimes_{k=2}^{N} X_{\delta(k,d)}\right), & \text{if } B = \Psi^+, \\ \left(\bigotimes_{n=2}^{N} \text{CNOT}_{\delta(1,d),\delta(n,d)}\right) H_{\delta(1,d)} \left(\bigotimes_{k=1}^{N} X_{\delta(k,d)}\right), & \text{if } B = \Psi^-, \end{cases} \tag{1}$$

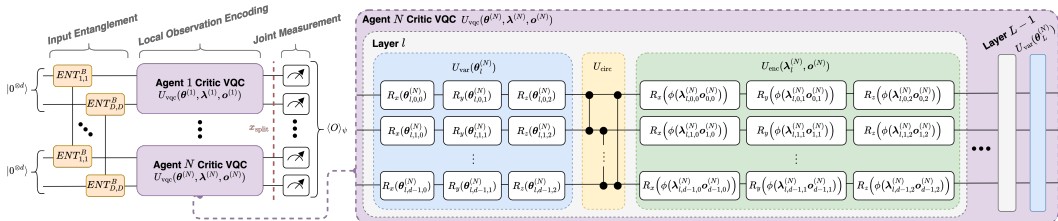

Figure 2: Quantum system design with $N$ agents and $D$ qubits per agent. Input entanglement operators (orange) couple the split critic VQCs (purple, with split point marked in red), which has cascaded layers of variational (blue), circular entanglement (yellow), and encoding (green) operators.

is a coupling operator across qubits $\{q_d^{(n)}\}_{n=1}^N \subseteq \mathcal{Q}$, where $\mathcal{B} \in \{\Phi^+, \Phi^-, \Psi^+, \Psi^-\}$ is the set of Bell states, $B \in \mathcal{B}$ is the selected entanglement scheme, and $\delta(n, d) = (n-1)D + d$ is an index mapping within $\mathcal{Q}$ for agent $n \in \mathcal{N}$ at qubit index $d \in [1, D]$, i.e., $\delta(n, d) \equiv q_d^{(n)} \in \mathcal{Q}^{(n)}$. Importantly, this operator can be applied to entangle any arbitrary set of qubits within the circuit. Note that in this work we assume the agents receive their entangled qubits in real-time via a trusted central source, i.e., a central server, however, they could be pre-generated and stored in quantum memory at the agent if desired. The quantum circuits that generate each $B \in \mathcal{B}$ are given in Appendix C, Fig. C.1.

## 2.2 DECENTRALIZED SPLIT CRITIC VQC DESIGN

At its core, our joint quantum critic is a split neural network (SNN) (Vepakomma et al., 2018), with each agent's local VQC serving as a branch. After the input qubits are entangled, they are partitioned back into $N$ sets of $D$ qubits, i.e., $\{\mathcal{Q}^{(n)}\}_{n \in \mathcal{N}}$, and transmitted to each agent respectively. The agents collect and encode local observations from the environment into their assigned qubits using a VQC. We use the VQC architectures of Jerbi et al. (2021); Skolik et al. (2021) for our hybrid quantum network design. Each agent in our QMARL setting uses the same VQC architecture for their branch of the critic, but tunes their own unique set of parameters. The same architecture is a reasonable assumption since all agents are learning in the same environment, and the uniqueness of parameters tailors each branch to local observations. From Fig. 2, the VQC design consists of $L$ cascaded layers of *variational*, *circular entanglement*, and *encoding* operators, with an additional variational layer at the end of the circuit before measurement. The trainable variational layer performs sequential parameterized Pauli X, Y, and Z-axis rotations, and it can be expressed as the unitary operator $U_{\text{var}}(\boldsymbol{\theta}_l^{(n)}) = \bigotimes_{d=0}^{D-1} R_z(\boldsymbol{\theta}_{l,d,2}^{(n)}) R_y(\boldsymbol{\theta}_{l,d,1}^{(n)}) R_x(\boldsymbol{\theta}_{l,d,0}^{(n)})$, where $\boldsymbol{\theta}^{(n)} \in [0, 2\pi]^{(L+1) \times D \times 3}$ is a matrix of rotation angle parameters for agent $n$. The non-trainable circular entanglement layer binds neighboring qubits within a single agent using the operator $U_{\text{circ}} = CZ_{0,D-1}\left(\prod_{d=0}^{D-2} CZ_{d,d+1}\right)$. The trainable encoding layer maps a matrix of classical features $\boldsymbol{o}^{(n)} \in \mathbb{R}^{D \times 3}$, i.e., an agent's environment observation, into a quantum state via the operator: $U_{\text{enc}}(\boldsymbol{\lambda}_l^{(n)}, \boldsymbol{o}^{(n)}) = \bigotimes_{d=0}^{D-1} R_z\big(\phi(\boldsymbol{\lambda}_{l,d,2}^{(n)} \boldsymbol{o}_{d,2}^{(n)})\big) R_y\big(\phi(\boldsymbol{\lambda}_{l,d,1}^{(n)} \boldsymbol{o}_{d,1}^{(n)})\big) R_x\big(\phi(\boldsymbol{\lambda}_{l,d,0}^{(n)} \boldsymbol{o}_{d,0}^{(n)})\big)$, where $\boldsymbol{\lambda}^{(n)} \in \mathbb{R}^{L \times D \times 3}$ is a matrix of trainable scaling parameters, and $\phi \colon \mathbb{R} \mapsto \mathbb{R}$ is an optional squash activation function. The entire VQC can be expressed as a single operator, as follows:

$$U_{\text{vqc}}(\boldsymbol{\theta}^{(n)}, \boldsymbol{\lambda}^{(n)}, \boldsymbol{o}^{(n)}) = U_{\text{var}}(\boldsymbol{\theta}_L^{(n)}) \prod_{l=0}^{L-1} U_{\text{enc}}(\boldsymbol{\lambda}_{L-l-1}^{(n)}, \boldsymbol{o}^{(n)}) \, U_{\text{circ}} \, U_{\text{var}}(\boldsymbol{\theta}_{L-l-1}^{(n)}), \quad (2)$$

which is parameterized by variational angles $\boldsymbol{\theta}^{(n)}$ and encoding weights $\boldsymbol{\lambda}^{(n)}$.

## 2.3 CENTRALIZED JOINT MEASUREMENT

The locally encoded qubits for each agent are subsequently forwarded to a central quantum server, which could either be the entanglement source or a different location, for joint measurement. A joint measurement across all qubits in the system is made in the Pauli $Z$ basis using the observable $O = \bigotimes_{d=1}^{N \times D} Z_d$. The joint value for the locally-encoded observations is then estimated as follows:

$$V(\boldsymbol{o}) \simeq w\left(\frac{1 + \langle O \rangle_\psi}{2}\right), \quad (3)$$

where $w \in \mathbb{R}$ is a learned scaling parameter, $\langle O \rangle_\psi$ is the expected value of the joint observable w.r.t. an arbitrary system state $|\psi\rangle$ across all qubits, and $\boldsymbol{o} = (\boldsymbol{o}^{(n)})_{n=1}^N$ is a vector of joint observations. This rescaling is necessary because the range of the measured observable is $\langle O \rangle_\psi \in [-1, 1]$ (i.e.,

---

**Algorithm 1:** Summary of eQMARL training using MAA2C for a quantum entangled split critic. The full algorithm is provided in Appendix D, Algorithm D.1

---

**Require:** Set of $N$ agents $\mathcal{N}$, 1 quantum entanglement source, 1 quantum measurement server

1: Initialize $N$ critic branches $U_{\text{vqc}}$ with $D$ qubits and parameters $\boldsymbol{\theta}_{\text{critic}}^{(n)}, \boldsymbol{\lambda}^{(n)}, \forall n \in \mathcal{N}$;
2: **for** *all episodes* **do**
3:      **for** *all time steps $\tau$* **do** ▷ eQMARL training, notation $\boldsymbol{o}_\tau = \left(\boldsymbol{o}_\tau^{(n)}\right)_{n=1}^N$, $\boldsymbol{o}_{\tau+1} = \left(\boldsymbol{o}_{\tau+1}^{(n)}\right)_{n=1}^N$,
         $\boldsymbol{a}_\tau = \left(a_\tau^{(n)}\right)_{n=1}^N$.
4:          Central quantum server generates $2N$ sets of $D$ entangled qubits and sends to agents via **quantum channel**;
5:          **for** *each agent $n \in \mathcal{N}$* **do**
6:              Apply $U_{\text{vqc}}(\boldsymbol{\theta}_{\text{critic}}^{(n)}, \boldsymbol{\lambda}^{(n)}, \boldsymbol{o}_\tau^{(n)})$ and $U_{\text{vqc}}(\boldsymbol{\theta}_{\text{critic}}^{(n)}, \boldsymbol{\lambda}^{(n)}, \boldsymbol{o}_{\tau+1}^{(n)})$ from (2) locally using assigned entangled input qubits;
7:              Transmit resulting qubits via **quantum channel**, and reward $r_\tau^{(n)}$ via **classical channel** to central quantum server;
8:          **end**
9:          Perform joint measurements on qubits across all agents to estimate $V(\boldsymbol{o}_\tau)$ and $V(\boldsymbol{o}_{\tau+1})$ using (3);
10:          Estimate $Q(\boldsymbol{o}_\tau, \boldsymbol{a}_\tau) = \sum_{n=1}^N r_\tau^{(n)} + \gamma V(\boldsymbol{o}_{\tau+1})$ using discount factor $\gamma$;
11:          Compute $\nabla_{x_{\text{split}}} \mathcal{L}_{\text{critic}}$ and transmit via **classical channel** to each agent to update $\boldsymbol{\theta}_{\text{critic}}^{(n)}$ locally using partial gradient from (4);
12:      **end**
13: **end**

---

proportional to the eigenvalues of the operator $O$), whereas $V(\boldsymbol{o}) \in \mathbb{R}$. The critic loss with respect to the joint value and local agent rewards is then disseminated amongst the agents for tuning of their localized portion of the split critic network and local policy networks.

## 2.4 SPLIT CRITIC LOSS

The loss of the split critic is derived in a way similar to Vepakomma et al. (2018). Since the input entanglement stage of eQMARL has no trainable parameters, it does not exist for the purposes of SL backpropagation. We denote the point of joint quantum measurement as the *split point*, which is preceded by local agent VQC *branches*. Each branch can be individually tuned using the partial gradient of the loss at the split point via partial gradient w.r.t. its own local parameters. If we define $x_{\text{split}}$ as the split point, then the partial gradient of each branch's parameters can be estimated using the central loss, as follows:

$$\nabla_{\boldsymbol{\theta}_{\text{critic}}^{(n)}} \mathcal{L}_{\text{critic}} = \frac{\partial \mathcal{L}_{\text{critic}}}{\partial \boldsymbol{\theta}_{\text{critic}}^{(n)}} = \frac{\partial \mathcal{L}_{\text{critic}}}{\partial x_{\text{split}}} \frac{\partial x_{\text{split}}}{\partial \boldsymbol{\theta}_{\text{critic}}^{(n)}} = \underbrace{\left(\nabla_{x_{\text{split}}} \mathcal{L}_{\text{critic}}\right)}_{\text{Central server}} \underbrace{\left(\nabla_{\boldsymbol{\theta}_{\text{critic}}^{(n)}} x_{\text{split}}\right)}_{\text{Local agent}}, \tag{4}$$

where $\nabla_{x_{\text{split}}} \mathcal{L}_{\text{critic}}$ is the gradient of the loss at the split point, and $\nabla_{\boldsymbol{\theta}_{\text{critic}}^{(n)}} x_{\text{split}}$ is the gradient from the split point back to the start of branch $n \in \mathcal{N}$. The value of $\nabla_{x_{\text{split}}} \mathcal{L}_{\text{critic}}$ is sent classically to the agents, and since (3) only uses a single trainable parameter, $w$, the classical communication overhead needed for split backpropagation is minimal. Here, we use the `Huber` loss for the critic (see Appendix E).

## 2.5 COUPLED AGENT LEARNING ALGORITHM

Our eQMARL uses a variation of the multi-agent advantage actor-critic (MAA2C) algorithm (Papoudakis et al., 2021) to train local agent policies with a split quantum joint critic. Here, we summarize the algorithm in Algorithm 1, which focuses on the elements for necessary for tuning the critic. In eQMARL, there are $N$ quantum agents that are physically separated from each other (no cross-agent communication is assumed) and one central quantum server. Each agent $n \in \mathcal{N}$ employs a VQC, given by (2), with unique parameters $\boldsymbol{\theta}_{\text{critic}}^{(n)}$ and $\boldsymbol{\lambda}^{(n)}$, that serves as one branch in the split critic network. All agents interact with the environment independently and each has its own local data buffer – local observations are neither shared amongst agents nor with the server. The first stage of eQMARL is fundamentally similar to traditional MAA2C. The second stage is where the uniqueness of eQMARL comes into play. The central quantum server prepares $2N$ sets of $D$ entangled qubits using (1) for each time step $\tau$, which are then transmitted to the agents via a quantum channel. Each agent then encodes their local observations $\boldsymbol{o}_\tau^{(n)}$ and $\boldsymbol{o}_{\tau+1}^{(n)}$ using (2) and their assigned entangled input qubits, and transmits the resulting qubits via quantum channel back to the server. The agents also share their corresponding reward $r_\tau^{(n)}$ via a classical channel with the server, which will be used for downstream loss calculations. Access to the reward is necessary for the critic to evaluate agent policy performance. This is a reasonable assumption in eQMARL as the reward value contains no localized environment information, and is also used in Yun et al. (2022a; 2023b); Park et al. (2023); regardless, the classical channel will also be used to transmit partial gradients of the critic loss. The server then performs a joint measurement on all the qubits associated with $\boldsymbol{o}_\tau^{(n)}$

and $\boldsymbol{o}_{\tau+t}^{(n)}$ to obtain estimates for $V(\boldsymbol{o}_\tau)$ and $V(\boldsymbol{o}_{\tau+1})$ using (3). Subsequently, the server computes the expected cumulative reward $Q(\boldsymbol{o}_\tau, \boldsymbol{a}_\tau)$ for the joint observations and actions at $\tau$ using $V(\boldsymbol{o}_{\tau+1})$, discount factor $\gamma$, and the respective rewards. The joint critic loss $\mathcal{L}_{\text{critic}}$ is then computed, its partial gradient w.r.t. the split point $\nabla_{x_{\text{split}}} \mathcal{L}_{\text{critic}}$ is estimated, and then sent via a classical channel to each agent to update their local weights $\boldsymbol{\theta}_{\text{critic}}^{(n)}$ using (4).

## 3 EXPERIMENTS AND DEMONSTRATIONS

### 3.1 ENVIRONMENTS

We use the `CoinGame` environment first proposed in Lerer & Peysakhovich (2018), and as implemented in Phan et al. (2022), which has been widely used (Foerster et al., 2018b; Phan et al., 2022; Kölle et al., 2024), a multi-agent variant of the canonical `CartPole` environment (Barto et al., 1983), and a multi-agent variant of the `MiniGrid` environment (Chevalier-Boisvert et al., 2023) as benchmarks for MARL scenarios. In particular, `CoinGame`'s nature as a zero-sum game and the independent natures of both multi-agent `CartPole` and `MiniGrid` serve as intriguing case studies for learning cooperative strategies using full, i.e., described by a Markov decision process (MDP), and partial, i.e., described by a partially observable MDP (POMDP), information. In `CoinGame`, we evaluate agents using the *Score* metric, which aggregates all agent undiscounted rewards over a single episode. In both `CartPole` and `MiniGrid` we evaluate agents using the *total reward* metric, which aggregates the number of time steps an agent maintains pole balance and the reward received for maze navigation over a single episode respectively. See Appendix F for environment details.

### 3.2 EXPERIMENT SETUP

We compare eQMARL against three baselines that are considered the current state-of-the-art configurations in actor-critic CTDE: 1) **Fully centralized CTDE (fCTDE)**, a classical configuration where the critic is a simple fully-connected NN located at a central server, like in Gupta et al. (2017); Foerster et al. (2018a), and requires agents to transmit their local observations to the server via a classical channel; 2) **Split CTDE (sCTDE)**, a classical configuration where the critic is a branching NN encoder spread across the agents, which is combined using a centralized NN based on Rashid et al. (2018) located at a central server, and requires agents to transmit intermediate NN activations via a classical channel; and 3) **Quantum fully centralized CTDE (qfCTDE)**, a quantum variant of fCTDE where the critic is located at a central server, as in Yun et al. (2022a; 2023b); Park et al. (2023), and agents transmit their local observations via a classical channel. These baselines were carefully chosen to convey how a quantum entangled split critic eliminates local environment observation sharing, while reducing classical communication overhead by leveraging the quantum channel, and minimizing centralized computational complexity. In our experiments we simplify the setup by using policy sharing across the agents, as done in Yun et al. (2023b) and Chen (2023). All classical models were built using `tensorflow`, the quantum models using `tensorflow-quantum`, and `cirq` for quantum simulations. For `CoinGame`, all models were trained for 3000 epochs, with $T = 50$ steps, $\gamma = 0.99$. For `CartPole`, all models were trained for 1000 epochs, with a maximum of 500 steps per episode. For `MiniGrid`, all models were trained for 1000 epochs, with a maximum of $T = 50$ steps per episode. All models use the `Adam` optimizer with varying learning rates. The quantum models use $D = 4$ qubits, $L = 5$ layers, and $\phi = $ `arctan` activation. The classical models use $h = 12$ hidden units for `CoinGame` and `CartPole`, and $h = 100$ for `MiniGrid`. See Appendices G to I for further details. We conduct all experiments on a high-performance computing cluster with 128 CPU cores and 256 GB of memory per node. The training time of `sCTDE` for `CoinGame` MDP is $\approx 5.5$ minutes, and for POMDP is $\approx 7.5$ minutes. In contrast, the training time of eQMARL is $\approx 3.5$ hours and $\approx 8.5$ hours for MDP and POMDP respectively; this is in line with many current QMARL works (Yun et al., 2022a; 2023b; 2022b; Chen, 2023; Park et al., 2023; Kölle et al., 2024), and is indicative of the known computational complexities of running quantum simulations on classical hardware.

### 3.3 COMPARING QUANTUM INPUT ENTANGLEMENT STYLES

The first set of experiments demonstrate the effectiveness of various input entanglement styles used in eQMARL approach. We run two separate experiments using the `CoinGame-2` environment using both MDP and POMDP dynamics. The score metric results for both dynamics are shown in Fig. 3.

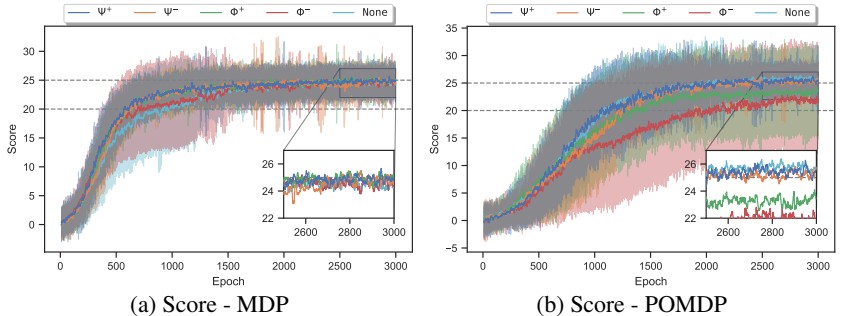

(a) Score - MDP                    (b) Score - POMDP

Figure 3: Comparison of `CoinGame-2` score performance with (a) MDP, and (b) POMDP dynamics for `eQMARL` using $\Psi^+$ (blue), $\Psi^-$ (orange), $\Phi^+$ (green), $\Phi^-$ (red), and `None` (cyan) entanglement averaged over 10 runs of 3,000 epochs, with $\pm 1$ std. dev. shown as shaded regions. These figures generally show that $\Psi^+$ outperforms other entanglement styles across both dynamics.

We consider score threshold markers of 20 and 25 to aid our discussion. In the MDP setting of Fig. 3a, we see see that $\Psi^+$ entanglement converges $4.5\%$ faster to a score threshold of 20 compared to the next closest $\Psi^-$. Similarly, $\Phi^+$ converges $5.2\%$ faster to a score of 25 compared to the next closest $\Psi^-$. At the end of training, all peak scores hover slightly above 25. Looking at the shaded standard deviation regions, we get a sense for the stability of each entanglement style. Specifically, we see that $\Psi^+$, $\Psi^-$, and $\Phi^+$ have similar tight ranges until epoch 1500, whereas both $\Phi^-$ and `None` have far lower minimum values until around epoch 1300. Moreover, $\Psi^-$ appears to have large downward spikes toward the end of training. Fig. 3a shows that there is a gap in convergence between $\Phi^-$ and `None`, and the other styles. Looking closer, we observe that $\Phi^+$ plateaus at earlier epochs, and $\Psi^-$ is more unstable (dropping in score) at later epochs. Hence, *we see a clear advantage for applying* $\Psi^+$. In the POMDP setting of Fig. 3b, we see that $\Psi^+$ converges $2\%$ faster to a score of 20 compared to the next closest `None`. Interestingly, there is a much larger gap in convergence between $\Phi^-$ and the others. A score of 25 is achieved $10.7\%$ faster by $\Psi^+$ compared to `None`, whereas both $\Phi^+$ and $\Phi^-$ never reach this threshold. The final peak scores for $\Psi^+$ and `None` hover slightly above 26. The shaded standard deviation regions exhibit a cascade effect between the styles, and, in particular, we observe that $\Phi^-$ has the lowest min, followed by $\Phi^+$ which has a slightly higher floor. These groupings are interesting as both $\Phi^+$ and $\Phi^-$ are similar in composition, only differing by a phase. Hence, *we again see a clear convergence and score advantage for using* $\Psi^+$.

Comparing the performance of the entanglement styles with both dynamics paints a picture of the generalizability of the system as a whole. Interestingly, the worse performance of $\Phi^+$ and $\Phi^-$ suggests that same-state entanglement, $|00\rangle$ and $|11\rangle$, regardless of phase, results in less coupling of agents compared to opposite-state entanglement, $|01\rangle$ and $|10\rangle$. One explanation for why the performance of `None` is similar to $\Psi^+$ in certain cases could be the degradation of fidelity (i.e., entanglement strength) within the agent VQCs. The circular entanglement unitary within an agent's VQC, defined as $U_{\text{circ}}$ from Section 2.2, binds the behavior of qubits within an agent by creating a "weakly" entangled state (i.e., low fidelity) from the preceding $U_{\text{var}}$. Introducing additional input entanglement could, in some cases, lower the fidelity of that entangled state further, resulting in poor model performance. It also could increase the fidelity, however, as similar to the process of entanglement distillation. We believe this decrease in fidelity is the reason why states like $\Phi^+$ and $\Phi^-$ perform poorly in most cases. Based on this, we also believe $\Psi^+$ performs better than $\Psi^-$ because of the decoherence associated with the difference in phase of the $|10\rangle$ term. Importantly, this polarization-dependent phase shift changes the structure of the entangled state entirely. Indeed, this phase is affected by the downstream $U_{\text{circ}}$, and thus results in entirely different measurement outcomes. The consistently high performance of $\Psi^+$ across both dynamics suggests that it enhances the generalizability of the system, and, since input entanglement does not increase classical computational overhead, we see that $\Psi^+$ entanglement can be used to couple the agents in both dynamics while achieving comparable or higher performance. Thus, *we select $\Psi^+$ as the entanglement scheme to be used in all subsequent experiments*.

### 3.4 CoinGame Experiments

We next compare the performance of `eQMARL-`$\Psi^+$ with baselines `fCTDE`, `sCTDE`, and `qfCTDE` using the `CoinGame-2` environment with MDP and POMDP state dynamics, as shown in Figs. 4a and 4b. Looking at the MDP score metric in Fig. 4a, we see that `eQMARL-`$\Psi^+$ converges $16.2\%$ *faster to a score threshold of 20* than the next-closest `qfCTDE`, and $10.8\%$ *faster to a score threshold*

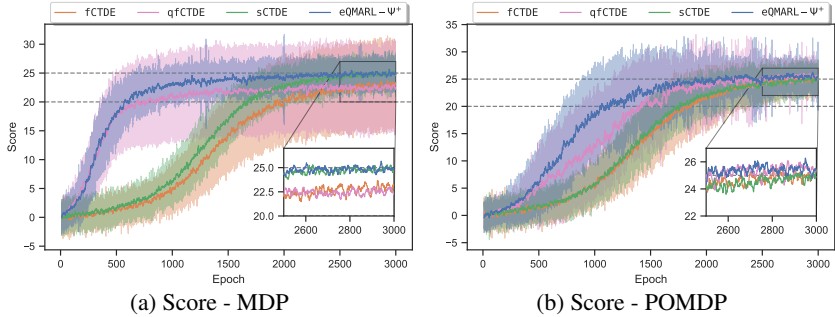

(a) Score - MDP                    (b) Score - POMDP

Figure 4: Comparison of `CoinGame-2` score performance with (a) MDP, and (b) POMDP dynamics for `fCTDE` (orange), `qfCTDE` (magenta), `sCTDE` (green), and `eQMARL`$-\Psi^+$ (blue) averaged over 10 runs of 3,000 epochs, with $\pm 1$ std. dev. shown as shaded regions. These figures generally show that eQMARL outperforms baselines across both environment dynamics.

*of 25* compared to `sCTDE`. Overall, we observe a 1.4% increase in max score for `eQMARL`$-\Psi^+$ compared to the next highest `sCTDE`. Additionally, `eQMARL`$-\Psi^+$ is smoother than `qfCTDE` at later epochs; suggesting that input entanglement stabilizes convergence. Fig. 4b shows that `eQMARL`$-\Psi^+$ converges 24% faster to a score of 20 considering the noticeable gap between it and `qfCTDE`. This demonstrates that the branching quantum network with input entanglement shortens convergence time compared to the fully centralized variant. Fig. 4b also shows that all models achieve a score of 25, however, in this case, `eQMARL`$-\Psi^+$ converges 17.8% faster and with slightly higher score than `qfCTDE`. Examining smoothness, we see much greater fluctuation between all curves compared to the MDP case. The difference in performance between the baselines in Figs. 4a and 4b demonstrates a clear quantum advantage for learning in the presence of partial information. Specifically, the faster convergence to the peak score threshold in `eQMARL`$-\Psi^+$ shows that splitting the quantum critic across the quantum channel with entangled input qubits allows the agents to learn a more cooperative strategy without direct access to local observations. This is interesting because `qfCTDE` is centralized and has all local observations at its disposal. This additional information would initially suggest better performance compared to approaches with only local information. However, from Fig. 4b, we observe a clear benefit for not only splitting the quantum critic as branches across the agents, but also introducing an entangled input state that couples their encoding behavior. Hence, from Figs. 4a and 4b, we conclude that *our proposed `eQMARL`$-\Psi^+$ configuration learns to play the game significantly faster than the classical variants* without sharing local observations, transmitting intermediate activations, nor tuning large NNs at the central server. The higher performance and shorter convergence time, compared to both the quantum and classical baselines, shows that splitting the critic amongst the agents results in no apparent loss in capability. In fact, the smoothness of the curves suggests that the input entanglement stabilizes the network over time.

## 3.5    CARTPOLE EXPERIMENTS

The next set of experiments compare the performance of eQMARL with baselines using a multi-agent variant of the `CartPole` environment with both MDP and POMDP state dynamics. The average reward metric across all agents in the environment for both dynamics is shown in Fig. 5. We use reward thresholds of the *mean* and *max* values to draw comparisons. From Fig. 5 we see that the classical models do not perform well overall in either setting, and `qfCTDE` experiences high variance in the MDP case. For MDP, `qfCTDE` achieves the highest mean and max rewards overall, but with an extremely high standard deviation. In contrast, `eQMARL`$-\Psi^+$ reaches its mean and max rewards 12.2% and 31.5% faster than `qfCTDE` respectively. For POMDP, we see that `sCTDE` achieves the highest overall max reward at the end of training, but with a very low mean. In contrast, both `qfCTDE` and `eQMARL`$-\Psi^+$ achieve a similar max value significantly earlier, about two-thirds of the way through training. `eQMARL`$-\Psi^+$ achieves the highest mean reward overall very early in training, which is 9.1% faster than `qfCTDE` with a similarly low standard deviation to the MDP setting. The key observation from the `CartPole` experiments is that `eQMARL`$-\Psi^+$ more rapidly learns a strategy with higher average reward than the classical variants in both MDP and POMDP settings. Further, `eQMARL`$-\Psi^+$ slightly outperforms `qfCTDE` in the POMDP setting with a higher average reward, and is much more stable with a significantly lower variance in the MDP setting – all achieved via implicit collaboration though entanglement. These results show that, without observation sharing,

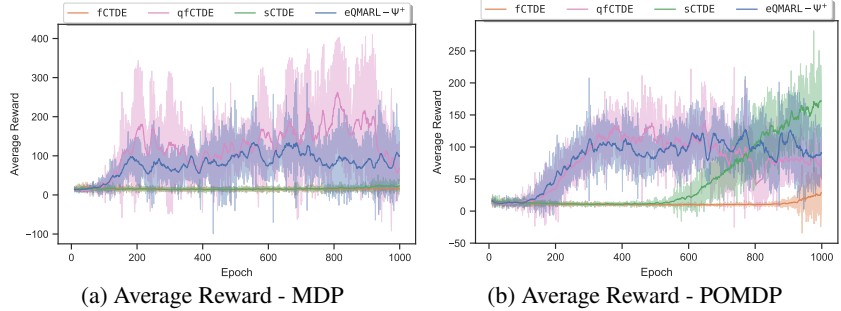

(a) Average Reward - MDP          (b) Average Reward - POMDP

Figure 5: Comparison of `CartPole` MDP and POMDP environment average reward performance for `fCTDE` (orange), `qfCTDE` (magenta), `sCTDE` (green), and `eQMARL`$-\Psi^+$ (blue) averaged over 5 runs of 1,000 epochs, with $\pm 1$ std. dev. shown as shaded regions. These figures generally show that eQMARL outperforms classical baselines and is more stable than `qfCTDE` across both dynamics.

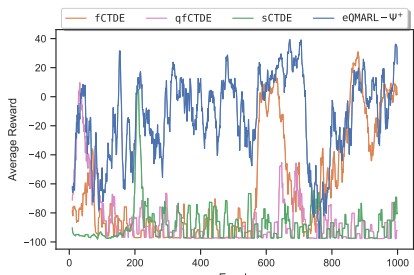

Figure 6: Comparison of `MiniGrid` POMDP environment reward performance for `fCTDE` (orange), `qfCTDE` (magenta), `sCTDE` (green), and `eQMARL`$-\Psi^+$ (blue) over 1000 epochs. This figure shows that eQMARL outperforms baselines by learning direct goal navigation instead of spinning in place.

`eQMARL`$-\Psi^+$ can learn a similarly performant, and dramatically more stable, strategy compared to a fully centralized quantum approach that has access to all agent observations.

## 3.6 MiniGrid Experiment

The next experiment compares the performance of eQMARL with baselines using a multi-agent variant of the `MiniGrid` environment with POMDP state dynamics, in which agents have a limited field of view. The average reward metric across all agents in the environment is shown in Fig. 6. From Fig. 6 we can see that, for the vast majority of training, the `fCTDE`, `qfCTDE`, and `sCTDE` baselines have an average reward that is clustered near $-100$; meaning that they exhaust many steps by simply spinning in place (since the maximum step size is 50, and $-2$ is the same-position reward). In contrast, we see that the average reward of `eQMARL`$-\Psi^+$ is spread out higher over the training regime with a mean of about $-13$, which is nearly 4.5-times higher than the other baselines. Indeed, this negative reward means that `eQMARL`$-\Psi^+$ also expends actions turning in place, but the fact the reward is so close to zero implies these events occur at a vastly reduced frequency than the baselines. In testing, `eQMARL`$-\Psi^+$ was able to traverse to the goal in as little as 9 steps, whereas `fCTDE` required 17 steps, and both `qfCTDE` and `sCTDE` were unable to find the goal within the 50 step limit. This is a marked 50% improvement in the exploration and navigation speed of `eQMARL`$-\Psi^+$ over `fCTDE`, with the bonus of no observation sharing. Hence, we have shown that `eQMARL`$-\Psi^+$ can indeed be applied to more complex environments, such as grid-world navigation with limited visibility, and provides learning benefits over baselines all without the need for observation sharing.

## 3.7 Ablation Study

The last set of experiments we consider is an ablation study to examine the relationship between NN layer depth and performance, and to facilitate fair model size comparisons. In particular, we trained `fCTDE` and `sCTDE` with hidden layer units $h \in \{3, 6, 12, 24\}$, and `qfCTDE` and `eQMARL` with VQC layers $L \in \{2, 5, 10\}$ on the `CoinGame-2` environment with MDP and POMDP state dynamics for 10 experiments of 3000 epochs each. An excerpt of the score metrics results for `eQMARL` and `sCTDE` in the MDP setting are shown in Fig. 7, and a comparison of the critic model sizes used in each framework is shown in Table 1. The full results are provided in Appendix I.5. In Fig. 7 we see that `eQMARL`$-\Psi^+$ with $L = 5$ achieves a mean score 3-times higher than $L = 2$, and nearly

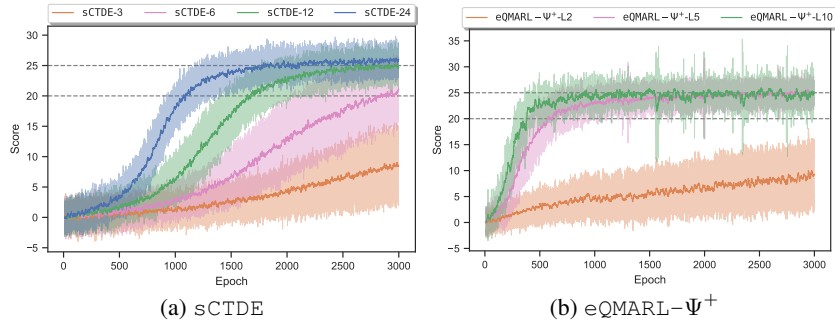

(a) sCTDE                    (b) eQMARL-$\Psi^+$

Figure 7: Ablation study score performance on MDP `CoinGame-2` for (a) sCTDE, and (b) eQMARL-$\Psi^+$ with hidden layer units $h \in \{3, 6, 12, 24\}$ and VQC layers $L \in \{2, 5, 10\}$, averaged over 10 runs of 3000 epochs, with $\pm 1$ std. dev. shown as shaded regions. These figures generally show that selecting parameters $h = 12$ and $L = 5$ results in optimal performance.

Table 1: Comparison of the best critic model size in number of trainable parameters for each framework used on the `CoinGame-2` environment with MDP and POMDP dynamics.

| Framework | Ablation Selection | Number of critic parameters: MDP | Number of critic parameters: POMDP |
|---|---|---|---|
| eQMARL | $L = 5$ | 265 (132 per agent, 1 central) | 817 (408 per agent, 1 central) |
| qfCTDE | $L = 5$ | 265 | 817 |
| fCTDE | $h = 12$ | 889 | 673 |
| sCTDE | $h = 12$ | 913 (444 per agent, 25 central) | 697 (336 per agent, 25 central) |

identical to $L = 10$. This trend is similar for `qfCTDE`. Both `sCTDE` and `fCTDE` also exhibit similar behavior for hidden units; that is the performance of $h = 12$ is nearly 2-times that of $h = 6$, and only marginally less than $h = 24$. Considering the significant performance drops and increased variation incurred by reducing, and the limited gains by increasing, both $h$ and $L$, the selection of $h = 12$ and $L = 5$ results in the most comparable performance across all baselines. Looking at Table 1, the critic sizes reported are for the entire system. This distinction is important since both `eQMARL` and `sCTDE` split the critic network across the agents the total size of the agent-specific network is a fraction of the total size. For MDP dynamics, we observe that the quantum models are 4 times smaller than the classical variants. For POMDP, we observe that the quantum models are slightly larger than their classical counterparts. This is because, in POMDP, the quantum models require a classical NN for dimensionality reduction at the input of each encoder. While the overall system size is larger, however, the number of central parameters is significantly reduced in the quantum cases – requiring only 1 parameter instead of 25. This is important for scaleability because the baselines implement a full NN at the central server and its size scales with $N$. In contrast, eQMARL has only a single trainable parameter tied to the measurement observable, which will remain fixed regardless of $N$.

## 4 CONCLUSION

In this paper we have proposed eQMARL, a novel quantum actor-critic framework for training decentralized policies using a split quantum critic with entangled input qubits and joint measurement. Spreading the critic across the agents via a quantum channel eliminates sharing local observations, minimizes the classical communication overhead from sending model parameters or intermediate NN activations, and reduces the centralized classical computational burden through optimization of a single quantum measurement observable parameter. We have shown that $\Psi^+$ input entanglement improves agent cooperation and system generalizability across both MDP and POMDP environments. For MDP, we have shown that eQMARL converges to a cooperative strategy $10.8\%$ faster and with a higher score compared to sCTDE. Likewise, for POMDP, we have shown that eQMARL converges to a cooperative strategy $17.8\%$ faster and with slightly higher score compared to qfCTDE. Further, we have also shown that eQMARL outperforms classical baselines and exhibits more stable performance than qfCTDE in independent environments. Lastly, we have shown that eQMARL requires 25-times fewer centralized parameters compared to sCTDE. One limitation of this work is the computational complexity of quantum simulations on classical hardware, which is an ongoing topic of research for noisy intermediate-scale quantum (NISQ) systems of many qubits. Indeed, many recent works on QMARL (Yun et al., 2022a; 2023b; 2022b; Chen, 2023; Park et al., 2023; Kölle et al., 2024) have similar hardware requirements to ours. Further, many recent works on quantum networks (Van Meter et al., 2022; Pettit et al., 2023; Lei et al., 2023; Azuma et al., 2023) propose methods for generating and storing entangled qubits, which can support the type of entanglement required in our system.

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

## A  RELATED WORKS

QMARL is a nascent field, with few works applying the quantum advantage to scenarios with multiple agents (Yun et al., 2022a; 2023b; 2022b; Chen, 2023; Park et al., 2023; Kölle et al., 2024). Further, the application of quantum to SL is even newer, with Yun et al. (2023a) being the only prior work. In Yun et al. (2022a) and Yun et al. (2023b), the authors propose a novel approach to QMARL that integrates CTDE and quantum state encoding of environment states. Specifically, they deploy actor-critic QRL with VQCs as the core for both actor and critic architectures. The QMARL agents each deploy a local quantum actor network and learn collectively in CTDE fashion using a centralized critic and unified experience replay buffer. In particular, the localized actors each operate independently from the others in the network and yet critically must also share their environment state experience with the collective via the shared replay buffer.

In Yun et al. (2022b), the authors propose the quantum meta MARL (QM2ARL) framework which uses a central meta Q-learning agent to train other local agents. In particular they define angle and pole training, a technique which tunes classical parameters to describe the Bloch sphere orientation of encoded quantum information. The first angle parameter training stage describes mapped Q-values as regions on the Bloch sphere, which are used to create the centralized meta agent. This meta agent is then subsequently used to train local agents by orienting pole parameters toward their specific environments.

In Chen (2023), the author proposes quantum asynchronous advantage actor-critic (QA3C) as a framework for training decentralized QRL agents, which leverages a *global shared memory* and *agent-specific memories* used in conjunction to train parallel agents. Akin to classical federated learning (FL), QA3C deploys an actor-critic network within each decentralized agent. The gradients of these local networks are periodically sent to the global shared memory for aggregation into a global model. The global model parameters are then broadcast back to the agents to update their local models.

The work in Park et al. (2023) proposes a QMARL approach for autonomous mobility cooperation using actor-critic networks with CTDE in NISQ environments with a shared replay buffer. In particular, they consider deployment with NISQ era limitations, the most notable being a low number of qubits. They show that CTDE-based QMARL can be deployed on near-term quantum hardware to coordinate robotic agents in smart factory environments. Critically, the coordination within their proposed system stems from the centralized critic network and shared replay buffer that has become synonymous with CTDE frameworks.

In Kölle et al. (2024), the authors propose a QMARL approach using evolutionary optimization with a VQC design based on quantum classification networks and agent policies implemented as independent VQC models with shared local information. Variations of the parameters for these policies are trained in *populations* to evaluate and compare performance. The authors assume fully observable environments in both cooperative and competitive settings, which is influenced via changes in the per-agent reward.

Finally, in Yun et al. (2023a), the authors propose a method for applying split learning to QML for traditional machine learning (ML) classification tasks. They deploy device-local quantum neural networks (QNNs) to predict local labels and features which are transmitted classically to a central server and transformed into true set of labels and localized gradients for each device.

The resounding theme in Yun et al. (2022a; 2023b; 2022b); Chen (2023); Park et al. (2023); Kölle et al. (2024); Yun et al. (2023a) is the use of independent agents or branches that communicate and learn through centralized classical means. No prior work, however, makes use of the quantum channel as a medium for system coupling or for multi-agent collaboration. Indeed, the quantum elements serve as drop-in replacements for classical NN counterparts, and, importantly, the quantum channel between agents and the potential for sharing entangled qubit states go largely under-utilized. Simply put, entanglement and the quantum channel are potentially useful untapped cooperative resources intrinsic to QMARL that have largely unknown benefits.

# B    MORE COMPREHENSIVE PRELIMINARIES

## B.1    QUANTUM MUTLI-AGENT REINFORCEMENT LEARNING

We consider a reinforcement learning (RL) setting with multiple agents in environments with both full and partial information. The dynamics of a system with full information is described by a Markov game with an underlying MDP with tuple $M_{\text{MDP}} = \langle \mathcal{N}, \mathcal{S}, \mathcal{A}, \mathcal{P}, \mathcal{R} \rangle$ where $\mathcal{N}$ is a set of $N$ agents, $\mathcal{S} = \{\mathcal{S}^{(n)}\}_{n \in \mathcal{N}}$ is the set of joint states across all agents, $\mathcal{A} = \{\mathcal{A}^{(n)}\}_{n \in \mathcal{N}}$ is the set of joint actions, $\mathcal{S}^{(n)}$ and $\mathcal{A}^{(n)}$ are the set of states and actions for agent $n$, $\mathcal{P}(s_{t+1}|s_t, a_t)$ is the state transition probability, and $\mathcal{R}(s_t, a_t) = \{r_t^{(n)}\}_{n \in \mathcal{N}} \in \mathbb{R}^N$ is the joint reward $\forall s_t \in \mathcal{S}, a_t \in \mathcal{A}$. The dynamics of a system with partial information is described by a Markov game with an underlying POMDP with tuple $M_{\text{POMDP}} = \langle \mathcal{N}, \mathcal{S}, \mathcal{A}, \mathcal{P}, \mathcal{R}, \Omega, \mathcal{O} \rangle$ where $\mathcal{N}, \mathcal{S}, \mathcal{A}, \mathcal{P}$ are the same as in $M_{\text{MDP}}$, however the full state of the environment $s_t$ at time $t$ is kept hidden from the players. Instead, at time $t$ the agents receive a local observation from the set of joint observations $\Omega = \{\Omega^{(n)}\}_{n \in \mathcal{N}}$, where $\Omega^{(n)}$ is the set of observations for agent $n$, with transition probability $\mathcal{O}(o_t|s_{t+1}, a_t), \forall o_t \in \Omega$, which is dependent on the hidden environment state after taking a joint action. We treat $M_{\text{MDP}}$ as a special case of $M_{\text{POMDP}}$ where $o_t^{(n)} = s_t^{(n)}$, that is the observations represent the full environment state information. Hereinafter, all notations will use $o_t^{(n)}$ in place of the local environment state for brevity to encompass all cases.

QMARL is the application of quantum computing to MARL. A popular approach in MARL is through actor-critic architectures, which tune policies, called *actors*, via an estimator for how good or bad the policy is at any given state of an environment, called a *critic*. To do this, the critic needs access to the local agent environment observations to estimate the value for a particular environment state. Current state of the art approaches follow the CTDE framework which deploys the critic on a central server and the actors across decentralized agents. Because the critic and the agents are physically separated, CTDE requires the agents to transmit their local observations to the server for the critic to estimate the joint value, thereby publicizing potentially private local observations. Quantum is often integrated as a drop-in replacement for classical NNs, called VQCs, within many MARL systems. These trainable quantum circuits tune the state of *qubits*, the quantum analog of classical bits, using unitary *gate* operations.

## B.2    QUANTUM COMPUTATION

### B.2.1    QUBIT STATES

A *qubit* is the quantum mechanical analog to the classical bit. The state of a qubit is represented as a 2-dimensional unit vector in complex Hilbert space $\mathcal{H} \in \mathbb{C}^2$. The computational basis is the set of states $\left\{|0\rangle = [1 \quad 0]^T, |1\rangle = [0 \quad 1]^T\right\}$ which forms a complete and orthonormal basis in $\mathcal{H}$ (meaning $\langle 0|1\rangle = \langle 1|0\rangle = 0$ and $\langle 0|0\rangle = \langle 1|1\rangle = 1$). All qubit states can be expressed as a linear combination of any complete and orthonormal basis, such as the computational basis, which is called *superposition*. We adopt Dirac notation to describe arbitrary qubit states $|\psi\rangle = [\alpha \quad \beta]^T = \alpha |0\rangle + \beta |1\rangle \in \mathcal{H}$ (called "ket psi") where $|\alpha|^2 + |\beta|^2 = 1$, their conjugate transpose $\langle \psi| = |\psi\rangle^\dagger = [\alpha^* \quad \beta^*] = \alpha^* \langle 0| + \beta^* \langle 1|$ (called "bra psi"), the inner product $\langle \psi_1|\psi_2\rangle = \alpha_1^* \alpha_2 + \beta_1^* \beta_2$, and the outer product $|\psi_1\rangle\langle\psi_2| =$

$\begin{bmatrix} \alpha_1\alpha_2^* & \alpha_1\beta_2^* \\ \beta_1\alpha_2^* & \beta_1\beta_2^* \end{bmatrix}$. Quantum systems with $D$ qubits can also be represented by extending the above notation using the Kronecker (tensor) product where $\mathcal{H} = \bigotimes_{d=0}^{D-1} \mathcal{H}_d = (\mathbb{C}^2)^{\otimes D}$ is the complex space of the system state $|\psi\rangle = \bigotimes_{d=0}^{D-1} |\psi_d\rangle$ for all $|\psi_D\rangle \in \mathcal{H}_D$. States that can be represented as either a single ket vector, or a sum of basis states are called *pure* states. For example, $|0\rangle$, $|1\rangle$, and $\frac{1}{\sqrt{2}}(|0\rangle + |1\rangle)$ are all pure states in $\mathcal{H}$.

### B.2.2 QUANTUM GATES

A quantum *gate* is an unitary operator (or matrix) $U$, such that $UU^\dagger = \mathbb{I}$, where $\mathbb{I}$ is the identity matrix, acting on the space $\mathcal{H}$ which maps between qubit states. Here, we use the single-qubit Pauli gates

$$X = \begin{bmatrix} 0 & 1 \\ 1 & 0 \end{bmatrix}, \; Y = \begin{bmatrix} 0 & -i \\ i & 0 \end{bmatrix}, \; Z = \begin{bmatrix} 1 & 0 \\ 0 & -1 \end{bmatrix}, \tag{B.1}$$

with their parameterized rotations

$$R_X(\theta) = e^{-i\frac{\theta}{2}X}, \; R_Y(\theta) = e^{-i\frac{\theta}{2}Y}, \; R_Z(\theta) = e^{-i\frac{\theta}{2}Z}, \tag{B.2}$$

where $\theta \in \mathbb{R}[0, 2\pi]$, the Hadamard gate

$$H = \frac{1}{\sqrt{2}} \begin{bmatrix} 1 & 1 \\ 1 & -1 \end{bmatrix}, \tag{B.3}$$

the 2-qubit controlled-$X$ ($CX$, also called CNOT) gate

$$CX_{1,2} = \text{CNOT}_{1,2} = \begin{bmatrix} \mathbb{I} & \mathbf{0} \\ \mathbf{0} & X \end{bmatrix}, \tag{B.4}$$

and the controlled-$Z$ ($CZ$) gate

$$CZ_{1,2} = \begin{bmatrix} \mathbb{I} & \mathbf{0} \\ \mathbf{0} & Z \end{bmatrix}, \tag{B.5}$$

which are both controlled by qubit 1 and target qubit 2, where $\mathbf{0}$ is a $2 \times 2$ square matrix of zeros.

### B.2.3 ENTANGLEMENT

Consider two arbitrary quantum systems $A$ and $B$, represented by Hilbert spaces $\mathcal{H}_A$ and $\mathcal{H}_B$. We can represent the Hilbert space of the combined system using the tensor product $\mathcal{H}_A \otimes \mathcal{H}_B$. If the quantum states of the two systems are $|\psi\rangle_A$ and $|\psi\rangle_B$, then the state of the combined system can be represented as $|\psi\rangle_A \otimes |\psi\rangle_B$. Quantum states that can be cleanly represented in this form, i.e., separated by tensor product, are said to be *separable*. Not all quantum states, however, are separable. For example, if we fix a set of basis states $\{|0\rangle_A, |1\rangle_A\} \in \mathcal{H}_A$ and $\{|0\rangle_B, |1\rangle_B\} \in \mathcal{H}_B$, then a general state in the space of $\mathcal{H}_A \otimes \mathcal{H}_B$ can be represented as $|\psi\rangle = \sum_{a \in \{0,1\}} \sum_{b \in \{0,1\}} c_{a,b} |a\rangle_A \otimes |b\rangle_B$, which is separable if there exists $c_{a,c} = c_a c_b, \forall a, b \in \{0,1\}$, producing the isolated states $|\psi\rangle_A = \sum_{a \in \{0,1\}} c_a |a\rangle_A$ and $|\psi\rangle_B = \sum_{b \in \{0,1\}} c_b |b\rangle_B$. If, however, there exists one $c_{a,c} \neq c_a c_b$, then the combined state is inseparable. In such cases, if a state is inseparable, it is said to be *entangled*.

The four Bell states $\mathcal{B} = \{|\Phi^+\rangle_{AB}, |\Phi^-\rangle_{AB}, |\Psi^+\rangle_{AB}, |\Psi^-\rangle_{AB}\}$ form a complete basis for two-qubit systems $\mathcal{H}_A \otimes \mathcal{H}_B$, and have the form:

$$|\Phi^+\rangle_{AB} = \frac{1}{\sqrt{2}} \left( |0\rangle_A |0\rangle_B + |1\rangle_A |1\rangle_B \right), \tag{B.6}$$

$$|\Phi^-\rangle_{AB} = \frac{1}{\sqrt{2}} \left( |0\rangle_A |0\rangle_B - |1\rangle_A |1\rangle_B \right), \tag{B.7}$$

$$|\Psi^+\rangle_{AB} = \frac{1}{\sqrt{2}} \left( |0\rangle_A |1\rangle_B + |1\rangle_A |0\rangle_B \right), \tag{B.8}$$

$$|\Psi^-\rangle_{AB} = \frac{1}{\sqrt{2}} \left( |0\rangle_A |1\rangle_B - |1\rangle_A |0\rangle_B \right). \tag{B.9}$$

Since it is impossible to separate the states of $\mathcal{B}$ into individual systems $\mathcal{H}_A$ and $\mathcal{H}_B$, the four Bell states are entangled. In particular, the Bell states are pure entangled states of the combined system $\mathcal{H}_A \otimes \mathcal{H}_B$, but cannot be separated into pure states of systems $\mathcal{H}_A$ and $\mathcal{H}_B$. Additionally, the four Bell states can be generated by quantum circuits using a combination of Pauli operators with a

constant $|0\rangle_A |0\rangle_B$ input state as follows:

$$\left|\Phi^+\right\rangle_{AB} = \text{CNOT}_{1,2} \left(H \otimes \mathbb{I}\right) |0\rangle_A |0\rangle_B, \tag{B.10}$$

$$\left|\Phi^-\right\rangle_{AB} = \text{CNOT}_{1,2} \left(H \otimes \mathbb{I}\right) \left(X \otimes \mathbb{I}\right) |0\rangle_A |0\rangle_B, \tag{B.11}$$

$$\left|\Psi^+\right\rangle_{AB} = \text{CNOT}_{1,2} \left(H \otimes \mathbb{I}\right) \left(\mathbb{I} \otimes X\right) |0\rangle_A |0\rangle_B, \tag{B.12}$$

$$\left|\Psi^-\right\rangle_{AB} = \text{CNOT}_{1,2} \left(H \otimes \mathbb{I}\right) \left(X \otimes X\right) |0\rangle_A |0\rangle_B \tag{B.13}$$

### B.2.4 PROJECTIVE MEASUREMENTS AND OBSERVABLES

A projective measurement is a Hermitian and unitary operator $O$, such that $O = O^\dagger$ and $OO^\dagger = \mathbb{I}$, called an *observable*. The outcomes of a measurement are defined by an observable's spectral decomposition

$$O = \sum_{m=0}^{M-1} \lambda_m P_m, \tag{B.14}$$

where $M = 2^n$ represents the number of measurement outcomes for $n$ qubits, and $m$ is a specific measurement outcome in terms of eigenvalues $\lambda_m$ and orthogonal projectors $P_m$ in the respective eigenspace. According to the *Born rule* Born (1926); Logiurato & Smerzi (2012); Masanes et al. (2019), the outcome of measuring an arbitrary state $|\psi\rangle$ will be one of the eigenvalues $\lambda_m$, and the state will be projected using the operator $P_m/\sqrt{p(m)}$ with probability:

$$p(m) = \langle\psi| P_m |\psi\rangle = \langle P_m\rangle_\psi \tag{B.15}$$

The expected value of the observable with respect to the arbitrary state $|\psi\rangle$ is given by:

$$\mathbb{E}_\psi[O] = \sum_{m=0}^{M-1} \lambda_m p(m) = \langle O\rangle_\psi \tag{B.16}$$

### B.2.5 COMMUTING OBSERVABLES

A set of observables $\{O_1, \ldots, O_K\}$ share a common eigenbasis (i.e., a common set of eigenvectors with unique eigenvalues) if

$$[O_i, O_j] = O_i O_j - O_j O_i = 0 \quad \forall i, j \in [1, K] \tag{B.17}$$

i.e., their pair-wise commutator is zero. In such cases the observables in the set are said to be *pair-wise commuting*, which in practice means that all observables in the set can be measured at the same time.

## C  JOINT INPUT ENTANGLEMENT CIRCUITS

We use a variation of Bell state entanglement to couple the input qubits of the agent VQC branches. Specifically, we entangle based on the set of four Bell states $\mathcal{B} \in \{\Phi^+, \Phi^-, \Psi^+, \Psi^-\}$, as outlined in (B.10–B.13), using the circuits as shown in Fig. C.1. The circuits in Fig. C.1 generate a quantum state across $D$ qubits, which has the combined Hilbert space $\mathcal{H}^{\otimes D}$. In each of the entangled operators, we select one qubit, $q_1$, to serve as the master control, and all others, $q_2, \ldots, q_D$, serve as targets. The control qubit functions identically to canonical Bell state entanglement. Here, we extend the gate operations that normally apply to the second qubit, in an $\mathcal{H} \otimes \mathcal{H}$ system, to all qubits in $\mathcal{H}^{\otimes D-1}$. The resulting state is an entangled pure state in $\mathcal{H}^{\otimes D}$.

## D  FULL ALGORITHM

The following algorithm is an expanded version of Algorithm 1, as shown in Algorithm D.1. In Algorithm D.1, we include all operations necessary for training both the agents and the split critic. In eQMARL, there is a set of $N$ quantum agents $\mathcal{N}$ that are physically separated from each other (no cross-agent communication is assumed) and one central quantum server. Each agent $n \in \mathcal{N}$ employs an actor policy network $\pi_{\boldsymbol{\theta}_{\text{actor}}^{(n)}}(a|\boldsymbol{o}_t^{(n)})$ (which can either classical or quantum in nature) with parameters $\boldsymbol{\theta}_{\text{actor}}^{(n)}$, and a VQC given by (2) with unique parameters $\boldsymbol{\theta}_{\text{critic}}^{(n)}$ and $\boldsymbol{\lambda}^{(n)}$ that serves as one branch in the split critic network. In our experiments we simplify the setup by using policy sharing across the agents, as done in Yun et al. (2023b) and Chen (2023); in other words, $\boldsymbol{\theta}_{\text{actor}}^{(n)} = \boldsymbol{\theta}_{\text{actor}}^{(k)} \ \forall n, k \in \mathcal{N}$. All agents interact with the environment independently and each has its own local data buffer $\mathcal{D}^{(n)}$ populated with local observations, actions, rewards, and next observations

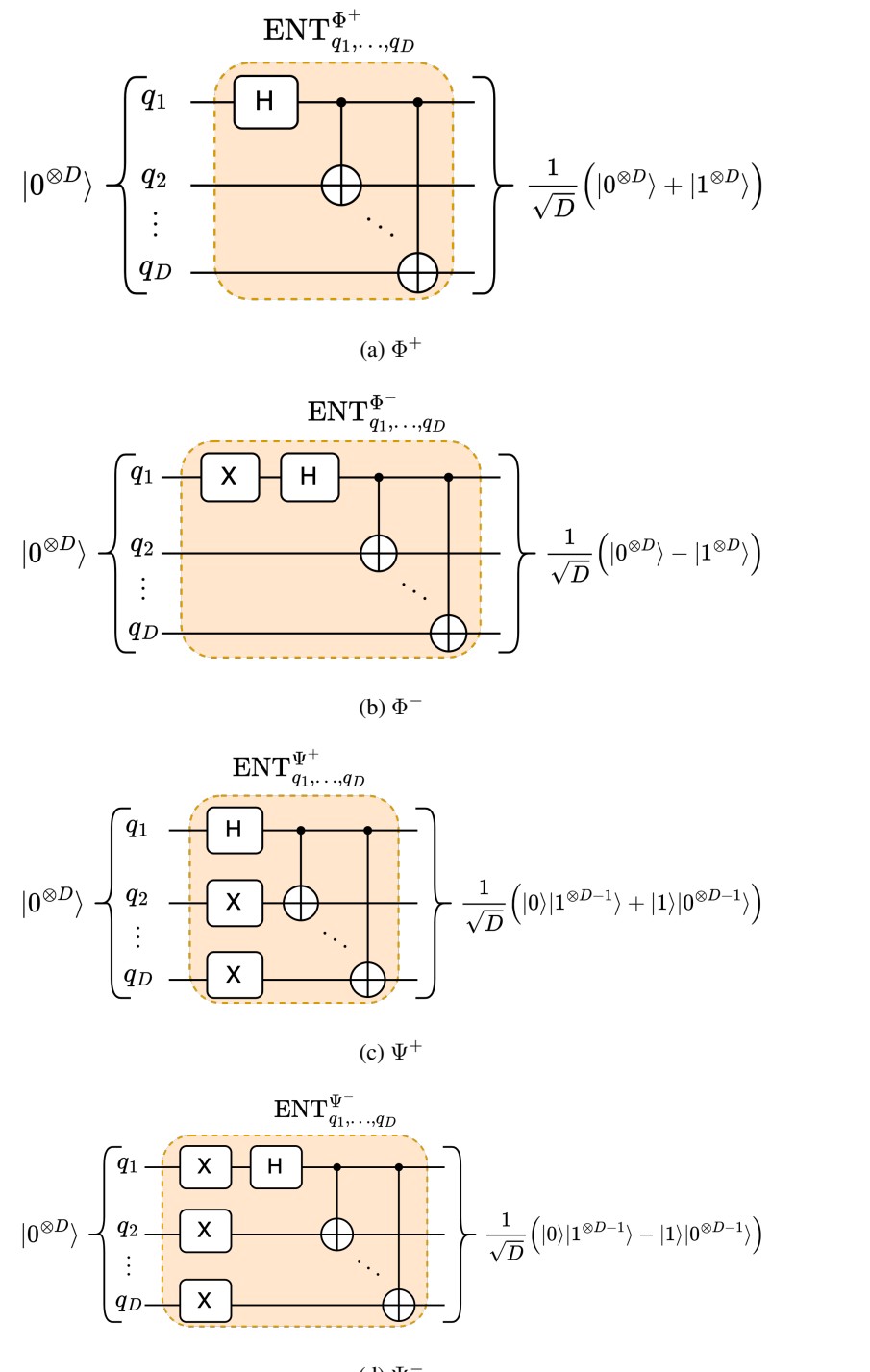

Figure C.1: Diagrams of joint entanglement operators based on the the four Bell states (a) $\Phi^+$, (b) $\Phi^-$, (c) $\Psi^+$, (d) $\Psi^-$.

represented by the 4-tuple $\left(\boldsymbol{o}_t^{(n)}, a_t^{(n)}, r_t^{(n)}, \boldsymbol{o}_{t+1}^{(n)}\right)$ for any instant in time $t$. These local data are not shared amongst agents, with only the reward and action being communicated classically to the central quantum server (the action only being necessary for policy sharing). Since we employ policy sharing, the final step in eQMARL with this in place is to also estimate the advantage value $A(\boldsymbol{o}_\tau, \boldsymbol{a}_\tau) = Q(\boldsymbol{o}_\tau, \boldsymbol{a}_\tau) - V(\boldsymbol{o}_\tau)$ using the existing value and expected reward estimates, compute actor loss $\mathcal{L}_{\text{actor}}$, compute the gradient of the loss w.r.t the shared actor parameters $\nabla_{\boldsymbol{\theta}_{\text{actor}}} \mathcal{L}_{\text{actor}}$, and

update $\boldsymbol{\theta}_{\text{actor}}$. Note that here we use `Huber` loss for the critic, and for the actors we use entropy-regularized advantage loss. The loss functions are described in detail in Appendix E.

---

**Algorithm D.1:** Full eQMARL training using MAA2C for a quantum entangled split critic.

---

**Require:** Set of $N$ agents $\mathcal{N}$, 1 quantum entanglement source, and 1 quantum measurement server

1: Initialize $N$ agent actor networks with shared parameters $\boldsymbol{\theta}_{\text{actor}}$ and local replay buffer $\mathcal{D}^{(n)} = \{\}, \forall n \in \mathcal{N}$;
2: Initialize $N$ critic branches $U_{\text{vqc}}$ with $D$ qubits and parameters $\boldsymbol{\theta}_{\text{critic}}^{(n)}, \boldsymbol{\lambda}^{(n)}, \forall n \in \mathcal{N}$;
3: **for** *episode=1, MaxEpisode* **do**
    ▷ `Localized environment interaction.`
4:     $t = 0$;
5:     $done = False$;
6:     **while** $done \neq True$ and $t < max\ steps$ **do**
7:         **for** *each agent* $n \in \mathcal{N}$ **do**
8:             Get local observation $\boldsymbol{o}_t^{(n)}$ from environment;
9:             Compute $\pi_{\boldsymbol{\theta}_{\text{actor}}^{(n)}}(a|\boldsymbol{o}_t^{(n)})$ and sample $a_t^{(n)}$;
10:            Apply action $a_t^{(n)}$ and get reward $r_t^{(n)}$ and next observation $\boldsymbol{o}_{t+1}^{(n)}$;
11:            Update local replay buffer $\mathcal{D}^{(n)} = \mathcal{D}^{(n)} \cup \left\{ \left( \boldsymbol{o}_t^{(n)}, a_t^{(n)}, r_t^{(n)}, \boldsymbol{o}_{t+1}^{(n)} \right) \right\}$;
12:            If $\boldsymbol{o}_{t+1}^{(n)}$ is terminal then communicate $done = True$;
13:         **end**
14:         $t = t + 1$;
15:     **end**
    ▷ `eQMARL framework for training.`
16:     **for** $\tau \in [0, t-2]$ **do**
17:         Central quantum server generates $2N$ sets of $D$ entangled qubits and transmits to agents via **quantum channel** (could be prepared a priori and stored in quantum memory);
18:         **for** *each agent* $n \in \mathcal{N}$ **do**
19:            Apply $U_{\text{vqc}}(\boldsymbol{\theta}_{\text{critic}}^{(n)}, \boldsymbol{\lambda}^{(n)}, \boldsymbol{o}_\tau^{(n)})$ and $U_{\text{vqc}}(\boldsymbol{\theta}_{\text{critic}}^{(n)}, \boldsymbol{\lambda}^{(n)}, \boldsymbol{o}_{\tau+1}^{(n)})$ from (2) locally using assigned entangled input qubits;
20:            Transmit via **quantum channel** the qubits after applying $U_{\text{vqc}}$ to central quantum server;
21:            Transmit via **classical channel** the reward $r_\tau^{(n)}$ and action $a_\tau^{(n)}$ at the current time step to central quantum server (only reward if policy sharing is not used);
22:         **end**
        ▷ `Using notation` $\boldsymbol{o}_\tau = \left(\boldsymbol{o}_\tau^{(n)}\right)_{n=1}^N$, $\boldsymbol{o}_{\tau+1} = \left(\boldsymbol{o}_{\tau+1}^{(n)}\right)_{n=1}^N$, $\boldsymbol{a}_\tau = \left(a_\tau^{(n)}\right)_{n=1}^N$.
23:         Perform joint measurements on qubits across all agents to estimate $V(\boldsymbol{o}_\tau)$ and $V(\boldsymbol{o}_{\tau+1})$ using (3);
24:         Estimate $Q(\boldsymbol{o}_\tau, \boldsymbol{a}_\tau) = \sum_{n=1}^N r_\tau^{(n)} + \gamma V(\boldsymbol{o}_{\tau+1})$ using discount factor $\gamma$;
25:         Estimate $A(\boldsymbol{o}_\tau, \boldsymbol{a}_\tau) = Q(\boldsymbol{o}_\tau, \boldsymbol{a}_\tau) - V(\boldsymbol{o}_\tau)$ for policy sharing;
26:         Compute $\nabla_{\boldsymbol{\theta}_{\text{actor}}} \mathcal{L}_{\text{actor}}$ and update $\boldsymbol{\theta}_{\text{actor}}$ for policy sharing;
27:         Compute $\nabla_{x_{\text{split}}} \mathcal{L}_{\text{critic}}$ and transmit via **classical channel** to each agent to update $\boldsymbol{\theta}_{\text{critic}}^{(n)}$ locally using partial gradient from (4);
28:     **end**
29: **end**

---

# E   LOSS FUNCTIONS

All of our actors and critics are trained using the same loss functions for each experiment. For the critics, we train using `Huber` loss

$$\mathcal{L}_{\text{critic}} = \frac{1}{T-1} \sum_{\tau=0}^{T-1} \begin{cases} \frac{1}{2}(V(\boldsymbol{o}_\tau) - Q(\boldsymbol{o}_\tau, \boldsymbol{a}_\tau))^2 & \text{if } V(\boldsymbol{o}_\tau) - Q(\boldsymbol{o}_\tau, \boldsymbol{a}_\tau) \leq \delta, \\ \delta |V(\boldsymbol{o}_\tau) - Q(\boldsymbol{o}_\tau, \boldsymbol{a}_\tau)| - \frac{1}{2}\delta^2 & \text{otherwise,} \end{cases} \quad \text{(E.1)}$$

where $\delta$ controls the point in which the loss function turns from quadratic to linear. In this work we use $\delta = 1$. For the actors, we deploy policy sharing amongst the agents. As such, all agents us the same policy parameters, and thus the loss must aggregate the individual losses of each agent. We train using the entropy-regularized advantage function

$$\mathcal{L}_{\text{actor}} = \frac{1}{n(T-1)} \sum_{n=1}^N \sum_{\tau=0}^{T-1} \left[ -A(o_\tau^{(n)}, a_\tau^{(n)}) \log_e p(a_\tau^{(n)}) + \alpha H(a_\tau^{(n)}) \right], \quad \text{(E.2)}$$

where
$$H(a_\tau^{(n)}) = -p(a_\tau^{(n)}) \log_e p(a_\tau^{(n)}), \tag{E.3}$$
is the entropy of selecting an action, $\alpha$ controls the influence of entropy, $n \in \mathcal{N}$ is the agent index, and $p(a_\tau^{(n)})$ is the probability of chosen action at time step $\tau$.

# F ENVIRONMENT SPECIFICATIONS

## F.1 COINGAME

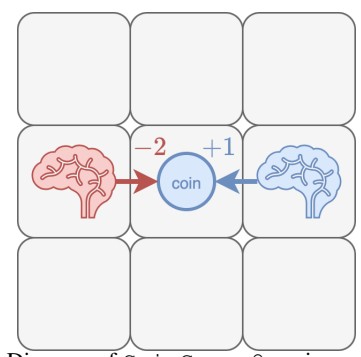

(a) Diagram of `CoinGame-2` environment.

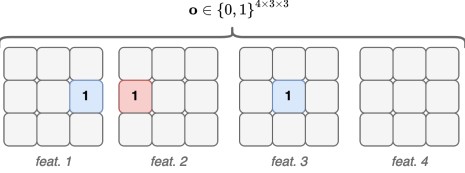

(b) MDP observation dynamics for blue player.

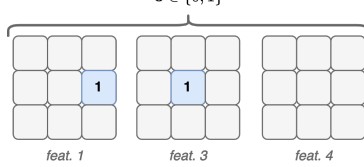

(c) POMDP observation dynamics for blue player.

Figure F.1: Example of (a) `CoinGame-2` environment with two players, colored red and blue, and a single coin colored blue, with visualization of observation matrix for the blue player with (b) MDP dynamics, and (c) POMDP dynamics. Grid squares in (b) and (c) colored grey denote a $0$ value, and colored blue/red squares denote a $1$ value.

In eQMARL, we train decentralized agents using the `CoinGame-2` environment first proposed in Lerer & Peysakhovich (2018), and as implemented in Phan et al. (2022). The `CoinGame-2` environment pits two agents of different colors (red and blue) on a $3 \times 3$ tile grid to collect coins with corresponding color. An example of `CoinGame-2` is shown in Fig. F.1a. Agent observations are a sparse matrix $\mathbf{o}^{(n)} \in \{0,1\}^{4\times3\times3} \ \forall n \in \mathcal{N}$ with 4 features each with a $3 \times 3$ grid world as shown in Fig. F.1b. The features specifically are: 1) A grid with a $1$ indicating the agent's location, 2) A grid with a $1$ to indicate other agent locations, 3) A grid with a $1$ for the location of coins that match the current agent's color, and 4) A grid with a $1$ for all other coins (different colors). Since these observations include all information about the game world, the game is considered fully observable and is described by an MDP. We also experiment with a partially observable variant of this game which removes the second feature from agent observations (i.e., the location of other agents), which is a space matrix $\mathbf{o}^{(n)} \in \{0,1\}^{3\times3\times3} \ \forall n \in \mathcal{N}$. In this partially observed setting, the game is described by a POMDP since agents cannot see each other and thus the full state of the game board is unknown. An example of this observation space is shown in Fig. F.1c. The agents can move along the grid by taking actions in the space $\mathcal{A}^{(n)} = \{\text{north, south, east, west}\} \ \forall n \in \mathcal{N}$. Each time an agent collects a coin of their corresponding color their episode reward is increased by $+1$, whereas a different color reduces their episode reward by $-2$. The goal for all agents is to maximize their discounted episode reward. The details of the environment are summarized in Table F.1.

We evaluate agents using three metrics: *score*, *total coins collected*, and *own coin rate*. The *score* metric aggregates all agent undiscounted rewards over a single episode
$$S = \sum_{n=1}^{N} \sum_{t=0}^{T-1} r_t^{(n)} \tag{F.1}$$
where $n \in \mathcal{N}$ is the agent index, $t \in [0, T-1]$ is the episode time index, $T$ is the episode time limit, and $r_t^{(n)}$ is the undiscounted agent reward at time $t$. The *total coins collected* metric gives insight into how active the agents were during the game
$$TC = \sum_{n=1}^{N} \sum_{t=0}^{T-1} c_t^{(n)} \tag{F.2}$$
where $c_t^{(n)}$ is the total number of coins collected by agent $n$ at time $t$. Finally, the *own coin rate* metric gives insight into how the agents achieve cooperation, specifically by being selective on which

Table F.1: Specifications for `CoinGame-2` environment with *MDP* and *POMDP* dynamics.

| Parameter | Value |
|---|---|
| Observation for agent $n$ at time $t$ | • MDP: $\boldsymbol{o}_t^{(n)} \in \{0,1\}^{4 \times 3 \times 3}$ (dimension is 36)
• POMDP: $\boldsymbol{o}_t^{(n)} \in \{0,1\}^{3 \times 3 \times 3}$ (dimension is 27) |
| Number of players ($N$) | 2 |
| Time limit ($T$) | 50 |
| Action for agent $n$ at time $t$ | $a_t^{(n)} \in \{\text{north, south, east, west}\}$ |
| Reward for agent $n$ at time $t$ | $r_t^{(n)} = \begin{cases} +1, & \text{if agent } n \text{ collects coin of same color,} \\ -2, & \text{if agent } n \text{ collects coin of different color,} \\ 0, & \text{otherwise} \end{cases}$ |
| Discount factor ($\gamma$) | 0.99 |
| Entropy coefficient ($\alpha$) | 0.001 |

coins they procure

$$OCR = \sum_{n=1}^{N} \sum_{t=0}^{T-1} k_t^{(n)} / c_t^{(n)} \tag{F.3}$$

where $k_t^{(n)}$ is the number of coins collected of the corresponding agent's color.

## F.2   CARTPOLE



Figure F.2: Example of an $N$-agent `CartPole` environment colored red, blue, and green.

We also train eQMARL using a multi-agent variant of the `CartPole` environment as proposed in Barto et al. (1983). The multi-agent `CartPole` environment runs multiple independent instances of the single-agent variant in parallel. This environment setup is an interesting case study for multi-agent learning because the observations of each agent are completely independent from one another; that is, observations from sibling environments are not strictly necessary to develop a strategy for a specific environment instance. This allows us to examine the impacts of both explicit and implicit cooperation between independent agents. An example of `CartPole` with $N$ agents is shown in Fig. F.2. Agent observations are a matrix $\mathbf{o}^{(n)} \in \mathbb{R}^{4 \times 1}$, $\forall n \in \mathcal{N}$ with 4 real-valued features. The features are: 1) Cart position with range $[-4.8, 4.8]$, 2) Cart velocity with range $(-\infty, \infty)$, 3) Pole angle in *radians* with range $[-0.418, 0.418]$, and 4) Pole angular velocity with range $(-\infty, \infty)$. The pole is considered *balanced* if the pole angle feature stays within the range $[-.2095, .2095]$ radians, and the cart position feature stays within the range $[-2.4, 2.4]$. These observations include all information about the environment, and thus the environment under these conditions is considered *fully observable* and described by an MDP. We also consider a *partially observed* variant of the environment which removes the second feature from agent observations (i.e., the cart velocity), which is a matrix $\mathbf{o}^{(n)} \in \mathbb{R}^{3 \times 1}$, $\forall n \in \mathcal{N}$. The environment is described by a POMDP in this setting since agents are unaware of their cart's velocity, and thus the full state of the environment is unknown. Notably, in this multi-agent variant of the environment the agent observations in both settings are independent from each other. The agents interact with the environment by taking actions in the space $\mathcal{A}^{(n)} = \{\text{left, right}\}$, $\forall n \in \mathcal{N}$, which correspond to pushing their cart to the left and right respectively. Similar to the observations, the agent actions are also independent and do not affect neighboring environments. Each time step an agent is successful in keeping their pole balanced they receive a $+1$ episode reward. The episode terminates when an observation falls outside of the balanced range. The goal for all agents is to maximize their expected total episode reward (i.e., the number of time steps they are able to keep the pole balanced). The details of the environment are summarized in Table F.2.

Table F.2: Specifications for multi-agent `CartPole` environment with *MDP* and *POMDP* dynamics.

| Parameter | Value |
|---|---|
| Observation for agent $n$ at time $t$ | • MDP: $\boldsymbol{o}_t^{(n)} \in \mathbb{R}^{4\times 1}$
• POMDP: $\boldsymbol{o}_t^{(n)} \in \mathbb{R}^{3\times 1}$ |
| Number of players ($N$) | 2 |
| Time limit ($T$) | 500 |
| Action for agent $n$ at time $t$ | $a_t^{(n)} \in \{\text{left}, \text{right}\}$ |
| Reward for agent $n$ at time $t$ | $r_t^{(n)} = \begin{cases} +1, & \text{if pole for agent } n \text{ is balanced,} \\ 0, & \text{otherwise} \end{cases}$ |

We evaluate the agents using the *average reward* metric, which aggregates all agent rewards over a single episode

$$AR = \frac{1}{N} \sum_{n=1}^{N} \sum_{t=0}^{T-1} r_t^{(n)} \tag{F.4}$$

where $n \in \mathcal{N}$ is the agent index, $t \in [0, T-1]$ is the episode time index, $T$ is the episode time limit, and $r_t^{(n)}$ is the agent reward at time $t$.

### F.3 MINIGRID

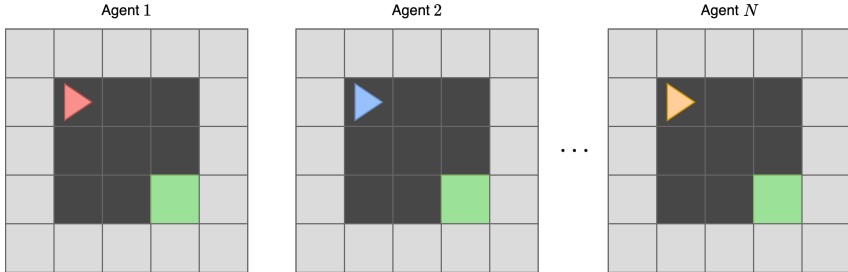

Figure F.3: Example of an $N$-agent `MiniGrid` environment colored red, blue, and orange.

We also train eQMARL using a multi-agent variant of the `MiniGrid` environment as proposed in Chevalier-Boisvert et al. (2023). The multi-agent `MiniGrid` environment runs multiple instances of the single-agent version in parallel. This environment configuration is an interesting case study for multi-agent learning because, similar to multi-agent `CartPole`, the local agent observations are independent of the others, and sharing observations is not necessary to solve the environment. This can, however, show how agent policies are affected with the seemingly added benefit of either directly shared or quantum-coupled local environment observations. An example of `MiniGrid` with $N$ agents is shown in Fig. F.3, and the details of the environment are summarized in Table F.3.

In the `MiniGrid` environment, the task is to find an optimal grid traversal path from a starting position to a goal using the action set $\mathcal{A}^{(n)} = \{\text{turn left}, \text{turn right}, \text{move forward}\}$, $\forall n \in \mathcal{N}$. Agent observations are a matrix $\mathbf{o}^{(n)} \in \mathbb{Z}^{7\times 7\times 3}$, $\forall n \in \mathcal{N}$, where the agent has a $7 \times 7$ limited field of view of the maze grid, and each cell in the grid is encoded as the 3-tuple $\langle \text{object}, \text{color}, \text{state} \rangle$, where $\text{object} \in [0, 10]$ identifies the object at the cell (e.g., empty, wall, etc.), $\text{color} \in [0, 5]$ identifies the color of the cell (e.g., red, green, etc.), and $\text{state} \in [0, 2]$ identifies the state of the cell (e.g., open, closed, and locked). We consider a $5 \times 5$ maze grid size in our experiments. Notably, because agent observations are a *limited field of view*, and not the full maze grid, we regard this environment as being described by POMDP dynamics. We use a reward shaping schedule of $-1$ for every step taken, $-2$ for standing still, 0 for not reaching the goal, and $100 \times (1 - 0.9 \times ((t+1)/T))$ for reaching the goal, where $t \in [0, T-1]$ is episode the time index, and $T$ is the episode time limit. Additionally, we provide the reward bonus $1/\sqrt{c}$ for actions that explore less visited grid positions and action pairs, where $c$ is the count for a specific grid position and action pair. In particular, there are four key factors that make this environment even more complex than the `CoinGame` and `CartPole` baselines. First, agents observe the environment using a limited field of view from their current position and rotational direction; hence, the agents must expend actions to both physically move and visually perceive the environment. Second, because the

Table F.3: Specifications for multi-agent `MiniGrid` environment with *POMDP* dynamics.

| Parameter | Value |
|---|---|
| Grid size | $w = 5, h = 5$ |
| Observation for agent $n$ at time $t$ | $\mathbf{o}_t^{(n)} \in \mathbb{Z}^{7 \times 7 \times 3}$ |
| Number of players ($N$) | 2 |
| Time limit ($T$) | 50 |
| Grid position for agent $n$ at time $t$ | $g_t^{(n)} \in [0, w \times h - 1]$ |
| Grid direction for agent $n$ at time $t$ | $d_t^{(n)} \in \{\texttt{up}, \texttt{down}, \texttt{left}, \texttt{right}\}$ |
| Action for agent $n$ at time $t$ | $a_t^{(n)} \in \{\texttt{turn left}, \texttt{turn right}, \texttt{move forward}\}$ |
| Reward for agent $n$ at time $t$ | $r_t^{(n)} = \frac{1}{\sqrt{c}} + \begin{cases} -2, & \text{if } g_t^{(n)} = g_{t-1}^{(n)}, \\ 100(1 - 0.9\frac{t+1}{T}) & \text{if goal is reached}, \\ -1, & \text{otherwise} \end{cases}$ $\text{where } c = \#\left(\langle g_t^{(n)}, d_t^{(n)}, a_t^{(n)} \rangle \in \{\langle g_i^{(n)}, d_i^{(n)}, a_i^{(n)} \rangle\}_{i=0}^{t-1}\right)$ |

field of view is limited, the goal position is not always in view; meaning that the agent strategies must also learn to search for the goal position in addition to finding an optimal traversal route. Third, the rotational direction of the agent plays a major role in both grid-world visibility and traversal actions; whereby an agent must learn to optimize the total number of rotation actions (i.e., `turn left` and `turn right`) for visual exploration and navigation – e.g., using a single `turn right` action (fewer steps) instead of three `turn left` actions (increased visibility). Fourth, agent actions are not limited when adjacent to grid-world `wall` positions; meaning that if an agent is facing `forward` to a `wall`, then the `move forward` action is still valid (even though clearly not optimal). Critically, the trade-off between increased visibility at the expense of rotation actions poses complex challenges here, and offers a unique opportunity for implicit observation sharing to improve navigation efficiency.

## G  QUANTUM ENCODING TRANSFORMATIONS

To encode environment observations into our quantum models we first apply a transformation on the observation matrix. This allows us to reduce its dimensions, thereby making it usable for the limited number of qubits available to NISQ systems, while also changing the range of matrix values to be suitable for input into one of the Pauli rotation gates.

### G.1  COINGAME-2 ENVIRONMENT

**MDP dynamics**  For the `CoinGame-2` environment with fully observed state dynamics we use the transformation
$$f_{\text{MDP}}(\mathbf{o}_{i \times j \times k}) = \sum_k \mathbf{o}_{i \times j, k} 2^{-k} \tag{G.1}$$
which sums over the last dimension of the observation matrix $\mathbf{o}_{i \times j \times k}$ with shape $i \times j \times k$. In the case of `CoinGame-2` with MDP dynamics the observations have shape $4 \times 3 \times 3$. This transformation reduces the dimensions to $4 \times 3 \times 1$, which can be directly fed into the encoder architecture outlined in Section 2.2.

**POMDP dynamics**  For the `CoinGame-2` environment with partially observed state dynamics our quantum models employ a small classical NN at the input of the encoder for dimensionality reduction, as done in Chen (2023). In particular, we use the transformation
$$f_{\text{POMDP}}(\mathbf{o}_{i \times j \times k}) = \mathbf{w}_{ijk \times 3d} \cdot \text{flatten}(\mathbf{o}_{i \times j \times k})^T + b \tag{G.2}$$
which flattens the observation matrix $\mathbf{o}_{i \times j \times k}$ with shape $i \times j \times k$ and passes it through a single fully-connected NN layer with parameters $\mathbf{w}_{ijk \times 3d}$ and $b$, and $d$ is the number of qubits. Note that, in POMDP, the trainable quantum encoding parameters $\boldsymbol{\lambda}^{(n)}$ are no longer necessary. In this case, we set $\boldsymbol{\lambda}^{(n)} = \mathbf{1}$, where $\mathbf{1}$ is a matrix of ones.

### G.2 CARTPOLE ENVIRONMENT

**Observation scaling**  For the `CartPole` environment we apply a constant observation scaling to both MDP and POMDP dynamics to normalize their values. In particular, we use the transformation

$$f(\mathbf{o}_{i \times j}) = \mathbf{o}_{i \times j} / \mathbf{v}_i \tag{G.3}$$

where

$$\mathbf{v} = [2.4, 2.5, 0.21, 2.5]^{\mathsf{T}} \tag{G.4}$$

is a constant scaling vector.

**POMDP dynamics**  For the `CartPole` environment with partially observed state dynamics we apply an additional transformation similar to the `CoinGame-2` POMDP case to reduce input feature dimensions. In particular, we apply the transformation,

$$f_{\text{POMDP}}(\mathbf{o}_{i \times j}) = \mathbf{w}_{ij \times 3d} \cdot \text{flatten}(\mathbf{o}_{i \times j})^T + b \tag{G.5}$$

which flattens the observation matrix $\mathbf{o}_{i \times j}$ with shape $i \times j$ and passes it through a single fully-connected NN layer with parameters $\mathbf{w}_{ij \times 3d}$ and $b$, and $d$ is the number of qubits.

### G.3 MINIGRID ENVIRONMENT

For the `MiniGrid` environment we apply a similar transformation to the `CoinGame` and `CartPole` POMDP environments, which is identical to (G.2). Specifically, we apply a fully-connected NN layer to reduce the dimensionality from the observation shape $7 \times 7 \times 3$ to $3d$, where $7 \times 7$ is the field of view of the agent as described in Table F.3, and $d$ is the number of qubits.

## H    MODEL HYPERPARAMETERS

The hyperparameters for each of the models trained in our experiments, as discussed in Section 3.2, are shown in Tables H.1 and H.2. Table H.1 show the model parameters used in qfCTDE and eQMARL. Table H.2 show the model parameters used in fCTDE and sCTDE.

Table H.1: Hyperparameters for qfCTDE and eQMARL, actor and critic, used on all environments.

| Environment | Model | Parameter | Value |
|---|---|---|---|
| CoinGame-2, CartPole | Actor, Critic | NN encoder transform activation | • MDP: N/A
• POMDP: linear |
| | | Flag $\boldsymbol{\lambda}^{(n)}$ as trainable | • MDP: True
• POMDP: False |
| | | Number of qubits per agent ($D$) | 4 |
| | | (eQMARL only) Input entanglement type ($B$) for critic | $\Psi^+$ |
| | | Number of layers ($L$) in $U_{\text{vqc}}$ | 5 |
| | | Squash activation ($\phi$) | arctan |
| | | Inverse temperature ($\beta$) | 1 |
| | | Optimizer | Adam |
| | | Learning rate | $[0.01, 0.1, 0.1]$ |
| MiniGrid | Critic | NN encoder transform activation | linear |
| | | Flag $\boldsymbol{\lambda}^{(n)}$ as trainable | False |
| | | Number of qubits per agent ($D$) | 4 |
| | | (eQMARL only) Input entanglement type ($B$) for critic | $\Psi^+$ |
| | | Number of layers ($L$) in $U_{\text{vqc}}$ | 5 |
| | | Squash activation ($\phi$) | arctan |
| | | Inverse temperature ($\beta$) | 1 |
| | | Optimizer | Adam |
| | | Learning rate | $[0.001, 0.001, 0.01, 0.1]$ |
| | Actor | Optimizer | Adam |
| | | Learning rate | $[0.0001]$ |
| | | Number of hidden units | 100 |

Table H.2: Hyperparameters for fCTDE and sCTDE, actor and critic, used on all environments.

| Environment | Model | Parameter | Value |
|---|---|---|---|
| CoinGame-2, CartPole | Actor, Critic | NN hidden units ($h$) | $[12]$ |
| | | Optimizer | Adam |
| | | Learning rate | 0.001 |
| MiniGrid | Actor, Critic | NN hidden units ($h$) | $[100]$ |
| | | Optimizer | Adam |
| | | Learning rate | 0.0001 |

# I    EXPERIMENT RESULTS

## I.1    ENTANGLEMENT STYLE COMPARISON

The empirical results for the entanglement comparison experiment, as discussed in Section 3.3, are shown in Tables I.1 and I.2. Table I.1 shows the score metric statistics mean, standard deviation, and $95\%$ confidence interval for each of the entanglement styles $\Psi^+$, $\Psi^-$, $\Phi^+$, $\Phi^-$, and None. Table I.2 shows the convergence time, in epochs, to each of the score thresholds 20, 25, and also to the maximum score value (reported parenthetically in *italics*) for each of the entanglement styles. The best values in each column are highlighted in **bold**.

Fig. I.1 shows the training results for the entanglement styles as discussed in Section 3.3. In particular, we provide the full set of performance metrics of score, total coins collected, own coins collected, and own coin rate, as outlined in Appendix F, (F.1–F.3). Fig. I.1 shows the results for the entanglement styles $\Psi^+$, $\Psi^-$, $\Phi^+$, $\Phi^-$, and None, as discussed in Section 3.3. The left column, Figs. I.1a, I.1c, I.1e, and I.1g, shows the performance for MDP environment dynamics. Similarly, the right column, Figs. I.1b, I.1d, I.1f, and I.1h, shows the performance for POMDP environment dynamics.

Table I.1: Comparison of entanglement style score performance for MDP and POMDP `CoinGame-2` environment dynamics using mean, standard deviation, and $95\%$ confidence interval statistics. Best values are highlighted in bold.

| Dynamics | Entanglement | Score | | |
|---|---|---|---|---|
| | | Mean | SD | 95% CI |
| MDP | $\Psi^+$ | **21.11** | 2.65 | (20.92, 21.29) |
| | $\Psi^-$ | 20.85 | 2.70 | (20.61, 21.07) |
| | $\Phi^+$ | 21.02 | **2.54** | (20.77, 21.30) |
| | $\Phi^-$ | 20.43 | 3.85 | (20.20, 20.59) |
| | None | 20.00 | 3.80 | (19.75, 20.20) |
| POMDP | $\Psi^+$ | 18.49 | **3.91** | (18.15, 18.74) |
| | $\Psi^-$ | 17.77 | 4.05 | (17.40, 18.09) |
| | $\Phi^+$ | 17.01 | 6.28 | (16.74, 17.25) |
| | $\Phi^-$ | 14.73 | 7.63 | (14.45, 15.01) |
| | None | **18.57** | 4.26 | (18.28, 18.82) |

Table I.2: Comparison of entanglement style score convergence (in number of epochs) for MDP and POMDP `CoinGame-2` environment dynamics. Best values are highlighted in bold.

| Dynamics | Entanglement | Epochs to Score Threshold | | |
|---|---|---|---|---|
| | | 20 | 25 | Max (*value*) |
| MDP | $\Psi^+$ | **568** | 2332 | 2942 (*25.67*) |
| | $\Psi^-$ | 595 | 1987 | 2849 (*25.45*) |
| | $\Phi^+$ | 612 | **1883** | 2851 (*25.51*) |
| | $\Phi^-$ | 691 | 2378 | 2984 (*25.23*) |
| | None | 839 | 2337 | **2495** (*25.12*) |
| POMDP | $\Psi^+$ | **1049** | **1745** | 2950 (*26.28*) |
| | $\Psi^-$ | 1206 | 2114 | 2999 (*25.95*) |
| | $\Phi^+$ | 1269 | - | 2992 (*24.1*) |
| | $\Phi^-$ | 1838 | - | 2727 (*22.8*) |
| | None | 1069 | 1955 | **2841** (*26.39*) |

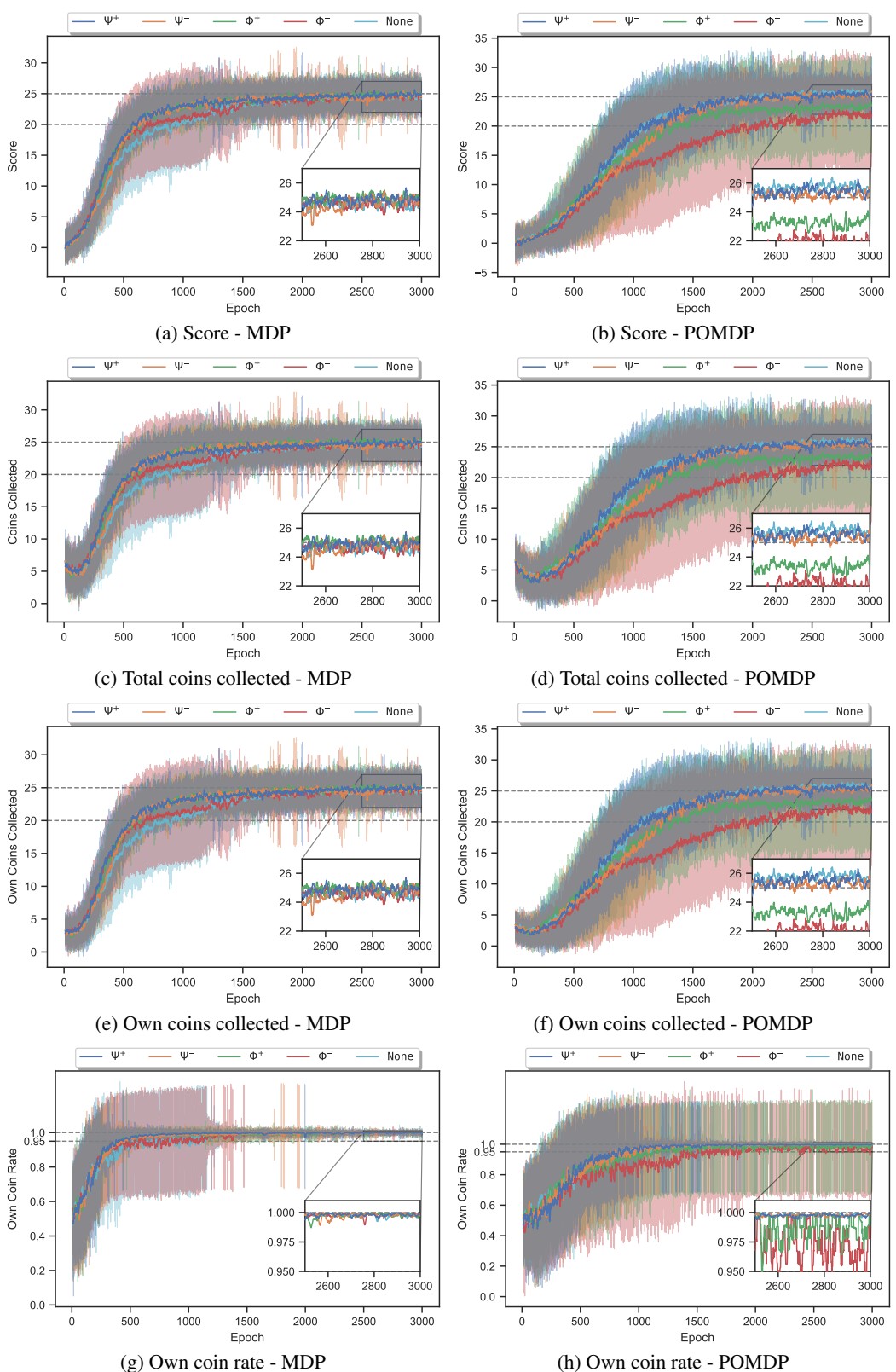

Figure I.1: Comparison of `CoinGame-2` MDP and POMDP environment performance metrics (a,b) score, (c,d) total coins collected, (e,f) own coins collected, and (g,h) own coin rate for eQMARL with varying input quantum entanglement styles $\Psi^+$ (blue), $\Psi^-$ (orange), $\Phi^+$ (green), $\Phi^-$ (red), and `None` (cyan) averaged over 10 runs, with $\pm 1$ std. dev. shown as shaded regions.

## I.2 COINGAME BASELINES COMPARISON

The empirical results for eQMARL-$\Psi^+$, qfCTDE, fCTDE, and sCTDE, as discussed in Section 3.4, are shown in Tables I.2 and I.4 and Fig. I.2. The performance for MDP dynamics is shown in Figs. I.2a, I.2c, I.2e, and I.2g, and for POMDP dynamics is shown in Figs. I.2b, I.2d, I.2f, and I.2h. Importantly, Fig. I.2 shed light on when, and how, a cooperative strategy is achieved by each framework. Further, through Fig. I.2 we also observe the relationship between the metrics outlined in Appendix F.1. This connection is important, as a single metric in isolation only paints part of the performance picture. A full comparison can be achieved by considering the metrics as as group, and, particularly, the relationship between agent score, i.e., the sum of rewards, and own coin rate, i.e., the priority given to coins of matching color.

Table I.3: Comparison of model score and own coin rate performance for MDP and POMDP `CoinGame-2` environment dynamics using mean, standard deviation, and $95\%$ confidence interval statistics. Best values are highlighted in bold.

| Dynamics | Framework | Score | | | Own Coin Rate | | |
|---|---|---|---|---|---|---|---|
| | | Mean | SD | 95% CI | Mean | SD | 95% CI |
| MDP | eQMARL-$\Psi^+$ | **21.11** | **2.65** | **(20.91, 21.37)** | **0.9640** | **0.0347** | **(0.9606, 0.9667)** |
| | qfCTDE | 19.41 | 6.23 | (19.22, 19.60) | 0.9398 | 0.1020 | (0.9367, 0.9423) |
| | sCTDE | 14.18 | 2.69 | (13.87, 14.53) | 0.8504 | 0.0928 | (0.8436, 0.8558) |
| | fCTDE | 12.36 | 4.41 | (12.01, 12.66) | 0.8202 | 0.1379 | (0.8153, 0.8255) |
| POMDP | eQMARL-$\Psi^+$ | **18.49** | 3.91 | **(18.24, 18.75)** | **0.9226** | 0.0831 | **(0.9173, 0.9281)** |
| | qfCTDE | 16.79 | 4.66 | (16.43, 17.19) | 0.9040 | 0.1135 | (0.8991, 0.9094) |
| | sCTDE | 13.70 | **2.79** | (13.33, 14.07) | 0.8466 | 0.0985 | (0.8407, 0.8525) |
| | fCTDE | 13.46 | 3.24 | (13.08, 13.75) | 0.8443 | 0.1026 | (0.8389, 0.8495) |

Table I.4: Comparison of model score and own coin rate convergence (in number of epochs) for MDP and POMDP `CoinGame-2` environment dynamics. Best values are highlighted in bold.

| Dynamics | Framework | Epochs to Score Threshold | | | Epochs to Own Coin Rate Threshold | | |
|---|---|---|---|---|---|---|---|
| | | 20 | 25 | Max (*value*) | 0.95 | 1.0 | Max (*value*) |
| MDP | eQMARL-$\Psi^+$ | **568** | **2332** | 2942 (*25.67*) | **376** | **2136** | **2136** (*1.0*) |
| | qfCTDE | 678 | - | **2378** (*23.38*) | 397 | - | 2832 (*0.9972*) |
| | sCTDE | 1640 | 2615 | 2631 (*25.3*) | 1511 | - | 2637 (*0.9864*) |
| | fCTDE | 1917 | - | 2925 (*23.67*) | 1700 | - | 2909 (*0.9857*) |
| POMDP | eQMARL-$\Psi^+$ | **1049** | **1745** | 2950 (*26.28*) | **773** | - | **2533** (*0.9997*) |
| | qfCTDE | 1382 | 2124 | 2871 (*26.09*) | 1038 | **2887** | 2887 (*1.0*) |
| | sCTDE | 1738 | 2750 | 2999 (*25.33*) | 1588 | - | 2956 (*0.9894*) |
| | fCTDE | 1798 | 2658 | **2824** (*25.49*) | 1574 | - | 2963 (*0.9894*) |

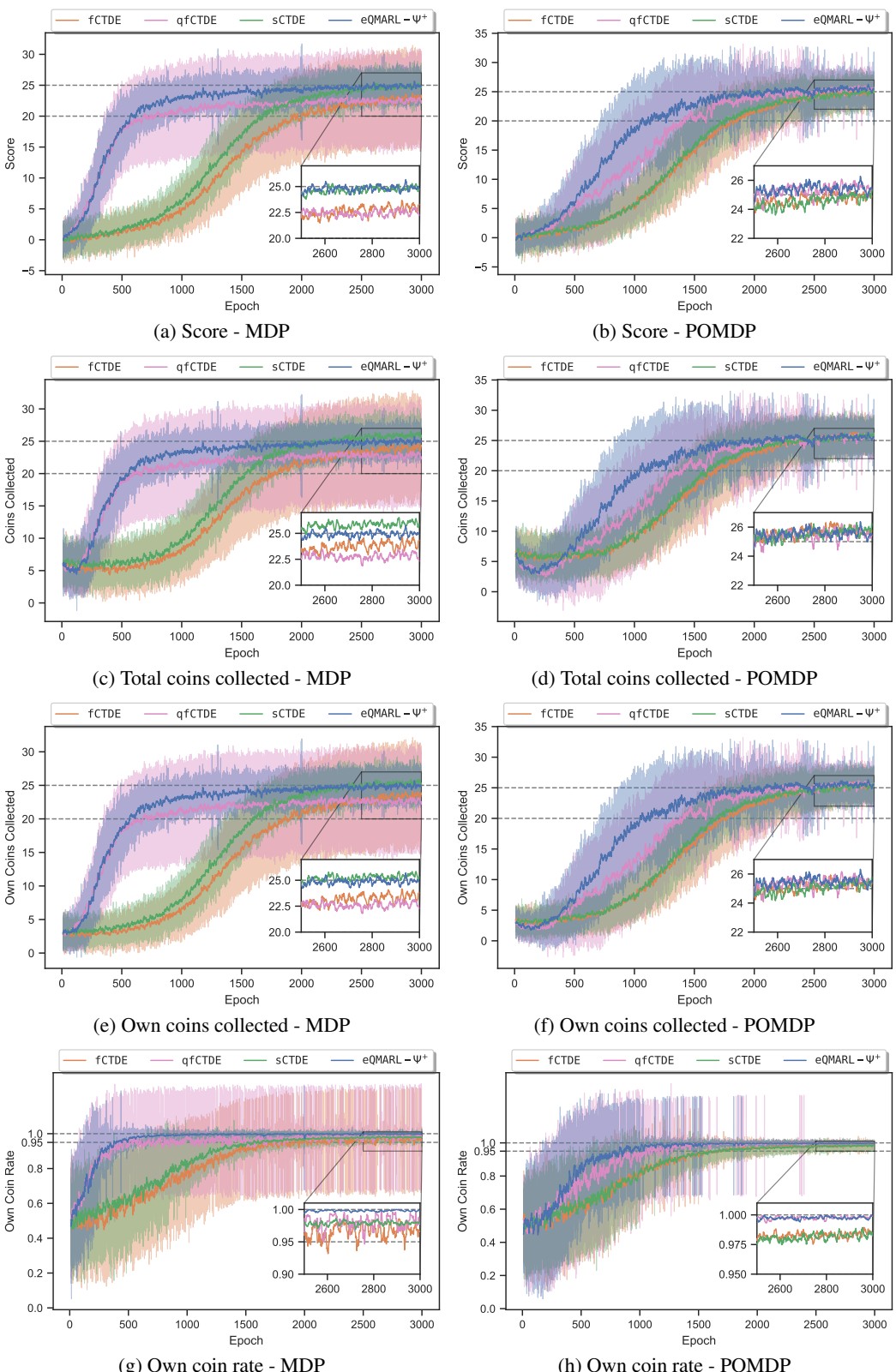

Figure I.2: Comparison of `CoinGame-2` MDP and POMDP environment performance metrics (a,b) score, (c,d) total coins collected, (e,f) own coins collected, and (g,h) own coin rate for `fCTDE` (orange), `qfCTDE` (magenta), `sCTDE` (green), and `eQMARL`-$\Psi^+$ (blue) averaged over 10 runs, with $\pm 1$ std. dev. shown as shaded regions.

## I.3 CARTPOLE BASELINES COMPARISON

The empirical results for eQMARL-$\Psi^+$, qfCTDE, fCTDE, and sCTDE, as discussed in Section 3.5, are shown in Tables I.5 and I.6 and Fig. I.3. The performance for MDP dynamics is shown in Fig. I.3a, and for POMDP dynamics is shown in Fig. I.3b. Importantly, from this we see that the classical models do not perform well overall in either setting, and qfCTDE experiences high variance in the MDP case. Even though sCTDE has a higher reward at the end of training in the POMDP case, it converges considerably more slowly, experiencing high variance at the end, and requires over 400 more epochs achieve a mean value less than half of eQMARL. In contrast, eQMARL is more stable than qfCTDE, and more rapidly converges to a higher mean reward than fCTDE and sCTDE across both settings.

Table I.5: Comparison of model average reward performance for MDP and POMDP CartPole environment dynamics using mean, standard deviation, and 95% confidence interval statistics.

| Dynamics | Framework | Reward | | |
| | | Mean | SD | 95% CI |
| --- | --- | --- | --- | --- |
| MDP | eQMARL-$\Psi^+$ | 79.11 | 50.62 | (77.26, 81.01) |
| | qfCTDE | 121.35 | 110.13 | (117.95, 124.59) |
| | sCTDE | 16.07 | 22.15 | (15.90, 16.21) |
| | fCTDE | 15.14 | 17.43 | (15.06, 15.22) |
| POMDP | eQMARL-$\Psi^+$ | 82.28 | 44.24 | (80.80, 83.91) |
| | qfCTDE | 79.03 | 44.06 | (76.27, 81.02) |
| | sCTDE | 47.59 | 29.71 | (44.71, 50.86) |
| | fCTDE | 11.62 | 32.02 | (11.45, 11.82) |

Table I.6: Comparison of model average reward convergence (in number of epochs) for MDP and POMDP CartPole environment dynamics.

| Dynamics | Framework | Epochs to Average Reward Threshold | |
| | | Mean (*value*) | Max (*value*) |
| --- | --- | --- | --- |
| MDP | eQMARL-$\Psi^+$ | 166 (*79.11*) | 555 (*134.16*) |
| | qfCTDE | 189 (121.35) | 810 (*262.43*) |
| | sCTDE | 23 (*16.07*) | 978 (*24.64*) |
| | fCTDE | 9 (*15.14*) | 44 (*19.43*) |
| POMDP | eQMARL-$\Psi^+$ | 251 (*82.28*) | 770 (*127.60*) |
| | qfCTDE | 276 (*79.03*) | 648 (*137.66*) |
| | sCTDE | 669 (*47.59*) | 998 (*172.16*) |
| | fCTDE | 9 (*11.62*) | 999 (*28.83*) |

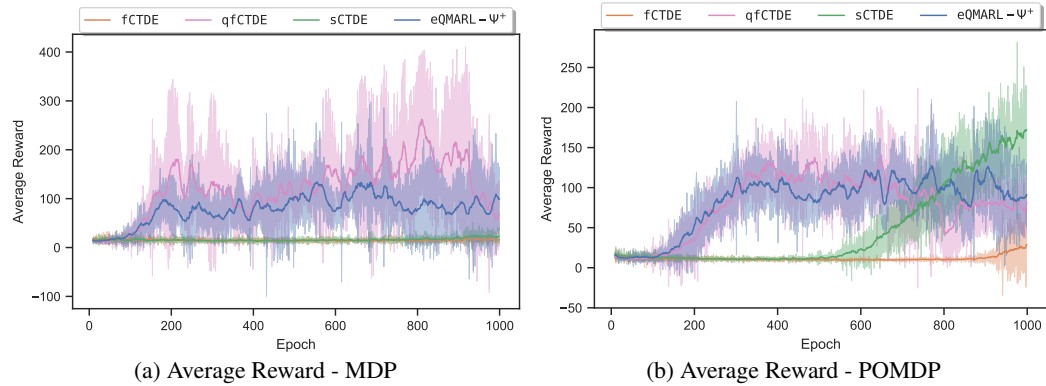

(a) Average Reward - MDP      (b) Average Reward - POMDP

Figure I.3: Comparison of CartPole MDP and POMDP environment average reward performance for fCTDE (orange), qfCTDE (magenta), sCTDE (green), and eQMARL-$\Psi^+$ (blue) averaged over 5 runs of 1000 epochs, with $\pm1$ std. dev. shown as shaded regions.

## I.4 MINIGRID BASELINES COMPARISON

The empirical results for eQMARL$-\Psi^+$, qfCTDE, fCTDE, and sCTDE, as discussed in Section 3.6, are shown in Table I.7 and Fig. I.4. Importantly, from Fig. I.4 we see that the qfCTDE, fCTDE, and sCTDE baselines have an average reward that is clustered near $-100$ for the majority of training. This implies that the baselines learn to exhaust many steps by simply spinning in place, since the maximum step size is 50 and the agents receive a $-2$ reward for staying in the same position as a previous time step. In contrast, we see that the average reward of eQMARL$-\Psi^+$ is spread out higher over the training regime. From Table I.7 we specifically see that eQMARL$-\Psi^+$ achieves an average overall reward of $-13.32$, which is 4.5-times higher than the baselines. Indeed, this negative reward means that eQMARL$-\Psi^+$ also expends actions turning in place, but the fact the reward is so close to zero implies these events occur at a vastly reduced frequency than the baselines.

In addition, from Table I.7 we see that both eQMARL$-\Psi^+$ and qfCTDE reduce the overall critic size by a factor of 8 compared to the classical baselines. This reduction in size means that eQMARL$-\Psi^+$ and qfCTDE are more computationally efficient than the classical baselines. Further, we see that eQMARL$-\Psi^+$ is even more efficient because it only requires a single centralized trainable parameter, which is a significant 200-times reduction in size compared to sCTDE.

In testing, eQMARL$-\Psi^+$ was able to traverse to the goal in as little as 9 steps, whereas fCTDE required 17 steps, and both qfCTDE and sCTDE were unable to find the goal within the 50 step limit. This is a marked 50% improvement in the exploration and navigation speed of eQMARL$-\Psi^+$ over fCTDE, with the bonus of no observation sharing, and an 8-times smaller overall critic size. Hence, we have shown that eQMARL$-\Psi^+$ can indeed be applied to more complex environments, such as grid-world navigation with limited visibility, and provide learning benefits over baselines without the need for observation sharing.

Table I.7: Comparison of model average reward performance for POMDP MiniGrid environment dynamics using mean and 95% confidence interval statistics, and comparison of model size in number of trainable critic parameters for each framework.

| Framework | Reward | | Number of Trainable Critic Parameters |
|---|---|---|---|
| | Mean | 95% CI | |
| fCTDE | -63.04 | (-65.16, -61.06) | 29,601 |
| qfCTDE | -85.86 | (-87.03, -84.72) | 3,697 |
| sCTDE | -88.02 | (-88.69, -87.10) | 29,801 (14,800 per agent, 201 central) |
| eQMARL$-\Psi^+$ | -13.32 | (-14.68, -11.91) | 3,697 (1,848 per agent, 1 central) |

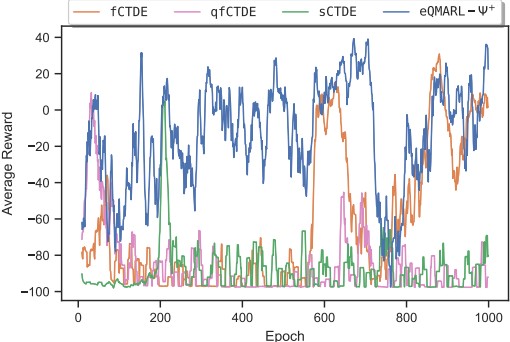

Figure I.4: Comparison of MiniGrid POMDP environment average reward performance for fCTDE (orange), qfCTDE (magenta), sCTDE (green), and eQMARL$-\Psi^+$ (blue) for 2 agents over 1000 epochs, with a maximum step limit of 50.

## I.5 ABLATION STUDY

The empirical results for the ablation study as discussed in Section 3.7 are shown in Table I.8 and Figs. I.5 and I.6. Looking at the statistics in Table I.8 and the convergence in Figs. I.5 and I.6, we can see that our selection of $h = 12$ hidden units for the baselines and $L = 5$ for the quantum models is fair because of the significant performance drops and increased variation incurred by reducing, and the limited gains by increasing, the number of units and layers. This choice for architecture results in the most comparable performance across all baselines. From the results of this ablation study, we can more concretely represent a comparison of the final sizes of the actor and critic models, in number of trainable parameters, for eQMARL, qfCTDE, fCTDE, and sCTDE. The final selected model sizes, in number of trainable parameters, are shown in Table I.9.

Table I.8: Ablation study with classical model hidden layer units $h \in \{3, 6, 12, 24\}$ and quantum VQC layers $L \in \{2, 5, 10\}$. Compares model size in number of trainable critic parameters with score and own coin rate performance for MDP and POMDP `CoinGame-2` environment dynamics using mean, standard deviation, and $95\%$ confidence interval statistics.

| Dynamics | Framework | Params | Score | | | Own Coin Rate | | |
| --- | --- | --- | --- | --- | --- | --- | --- | --- |
| | | | Mean | SD | 95% CI | Mean | SD | 95% CI |
| MDP | fCTDE-3 | 223 | 2.42 | 2.35 | (2.35, 2.49) | 0.6720 | 0.2024 | (0.6685, 0.6769) |
| | fCTDE-6 | 445 | 7.41 | 3.46 | (7.19, 7.65) | 0.7658 | 0.1414 | (0.7610, 0.7712) |
| | fCTDE-12 | 889 | 12.36 | 4.41 | (12.09, 12.67) | 0.8202 | 0.1379 | (0.8139, 0.8262) |
| | fCTDE-24 | 1777 | 17.63 | 2.58 | (17.25, 17.91) | 0.8823 | 0.0751 | (0.8770, 0.8875) |
| | sCTDE-3 | 229 | 3.24 | 3.09 | (3.16, 3.33) | 0.6852 | 0.1991 | (0.6821, 0.6897) |
| | sCTDE-6 | 457 | 8.54 | 3.67 | (8.29, 8.78) | 0.7857 | 0.1327 | (0.7804, 0.7924) |
| | sCTDE-12 | 913 | 14.18 | 2.69 | (13.90, 14.60) | 0.8504 | 0.0928 | (0.8454, 0.8553) |
| | sCTDE-24 | 1825 | 18.18 | 2.41 | (17.84, 18.53) | 0.8936 | 0.0673 | (0.8896, 0.8979) |
| | qfCTDE-L2 | 121 | 6.58 | 3.92 | (6.47, 6.66) | 0.8482 | 0.1921 | (0.8435, 0.8518) |
| | qfCTDE-L5 | 265 | 19.41 | 6.23 | (19.23, 19.59) | 0.9398 | 0.1020 | (0.9366, 0.9426) |
| | qfCTDE-L10 | 505 | 22.08 | 2.22 | (21.91, 22.26) | 0.9691 | 0.0247 | (0.9665, 0.9723) |
| | eQMARL-$\Psi^+$-L2 | 121 | 5.38 | 3.74 | (5.30, 5.46) | 0.8271 | 0.2213 | (0.8234, 0.8300) |
| | eQMARL-$\Psi^+$-L5 | 265 | 21.11 | 2.65 | (20.92, 21.35) | 0.9640 | 0.0347 | (0.9601, 0.9667) |
| | eQMARL-$\Psi^+$-L10 | 505 | 22.45 | 2.23 | (22.28, 22.62) | 0.9719 | 0.0219 | (0.9685, 0.9745) |
| POMDP | fCTDE-3 | 169 | 2.98 | 2.47 | (2.91, 3.05) | 0.7082 | 0.1890 | (0.7039, 0.7123) |
| | fCTDE-6 | 337 | 7.15 | 3.06 | (6.95, 7.37) | 0.7711 | 0.1388 | (0.7658, 0.7781) |
| | fCTDE-12 | 673 | 13.46 | 3.24 | (13.09, 13.76) | 0.8443 | 0.1026 | (0.8396, 0.8506) |
| | fCTDE-24 | 1345 | 17.38 | 2.65 | (17.06, 17.73) | 0.8889 | 0.0752 | (0.8840, 0.8945) |
| | sCTDE-3 | 175 | 2.68 | 2.60 | (2.61, 2.74) | 0.6834 | 0.1942 | (0.6792, 0.6866) |
| | sCTDE-6 | 349 | 6.35 | 3.53 | (6.18, 6.54) | 0.7677 | 0.1488 | (0.7633, 0.7725) |
| | sCTDE-12 | 697 | 13.70 | 2.79 | (13.44, 13.99) | 0.8466 | 0.0985 | (0.8411, 0.8515) |
| | sCTDE-24 | 1393 | 17.97 | 2.60 | (17.67, 18.25) | 0.8948 | 0.0723 | (0.8898, 0.9004) |
| | qfCTDE-L2 | 745 | 12.34 | 7.56 | (12.09, 12.60) | 0.8335 | 0.2058 | (0.8277, 0.8386) |
| | qfCTDE-L5 | 817 | 16.79 | 4.66 | (16.45, 17.04) | 0.9040 | 0.1135 | (0.8994, 0.9091) |
| | qfCTDE-L10 | 937 | 18.14 | 4.28 | (17.83, 18.31) | 0.9476 | 0.0660 | (0.9443, 0.9508) |
| | eQMARL-$\Psi^+$-L2 | 745 | 17.14 | 3.98 | (16.77, 17.47) | 0.8834 | 0.1106 | (0.8769, 0.8896) |
| | eQMARL-$\Psi^+$-L5 | 817 | 18.49 | 3.91 | (18.23, 18.80) | 0.9226 | 0.0831 | (0.9172, 0.9272) |
| | eQMARL-$\Psi^+$-L10 | 937 | 19.09 | 3.44 | (18.86, 19.46) | 0.9485 | 0.0603 | (0.9458, 0.9523) |

Table I.9: Comparison of the best model size in number of trainable parameters for each framework used on `CoinGame-2` environment with *MDP* and *POMDP* dynamics.

| Framework | Ablation Selection | Model | Number of Trainable Parameters | |
| --- | --- | --- | --- | --- |
| | | | *MDP dynamics* | *POMDP dynamics* |
| eQMARL | $L = 5$ | Actor | 136 | 412 |
| | $L = 5$ | Critic | 265 (132 per agent, 1 central) | 817 (408 per agent, 1 central) |
| qfCTDE | $L = 5$ | Actor | 136 | 412 |
| | $L = 5$ | Critic | 265 | 817 |
| fCTDE | $h = 12$ | Actor | 496 | 388 |
| | $h = 12$ | Critic | 889 | 673 |
| sCTDE | $h = 12$ | Actor | 496 | 388 |
| | $h = 12$ | Critic | 913 (444 per agent, 25 central) | 697 (336 per agent, 25 central) |

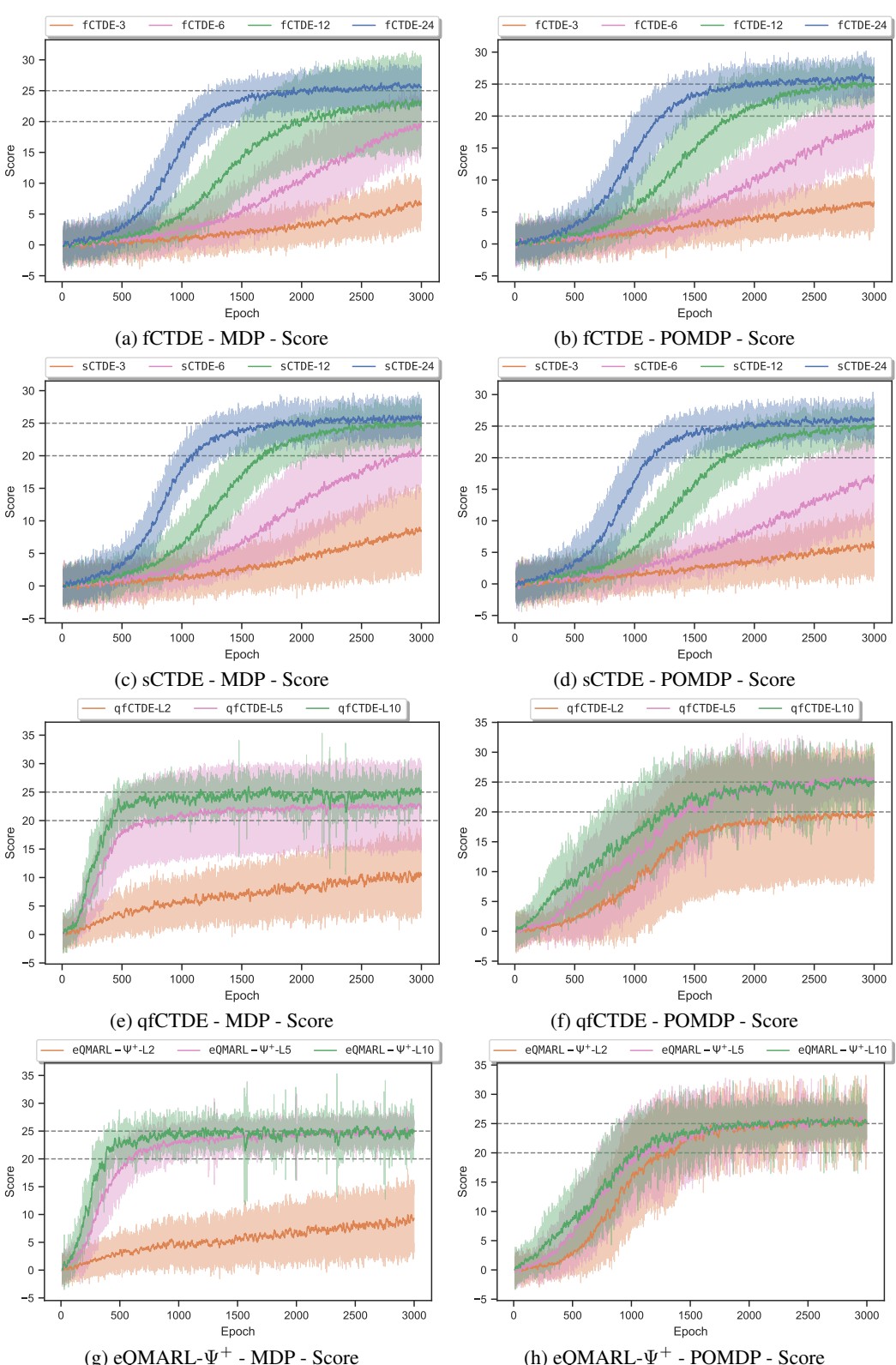

Figure I.5: Score performance for ablation study using `CoinGame-2` for (a,b) fCTDE, and (c,d) sCTDE, and (e,f) qfCTDE, and (g,h) eQMARL-$\Psi^+$ with hidden layer units $h \in \{3, 6, 12, 24\}$ and VQC layers $L \in \{2, 5, 10\}$, averaged over 10 runs of 3000 epochs, with $\pm 1$ std. dev. shown as shaded regions.

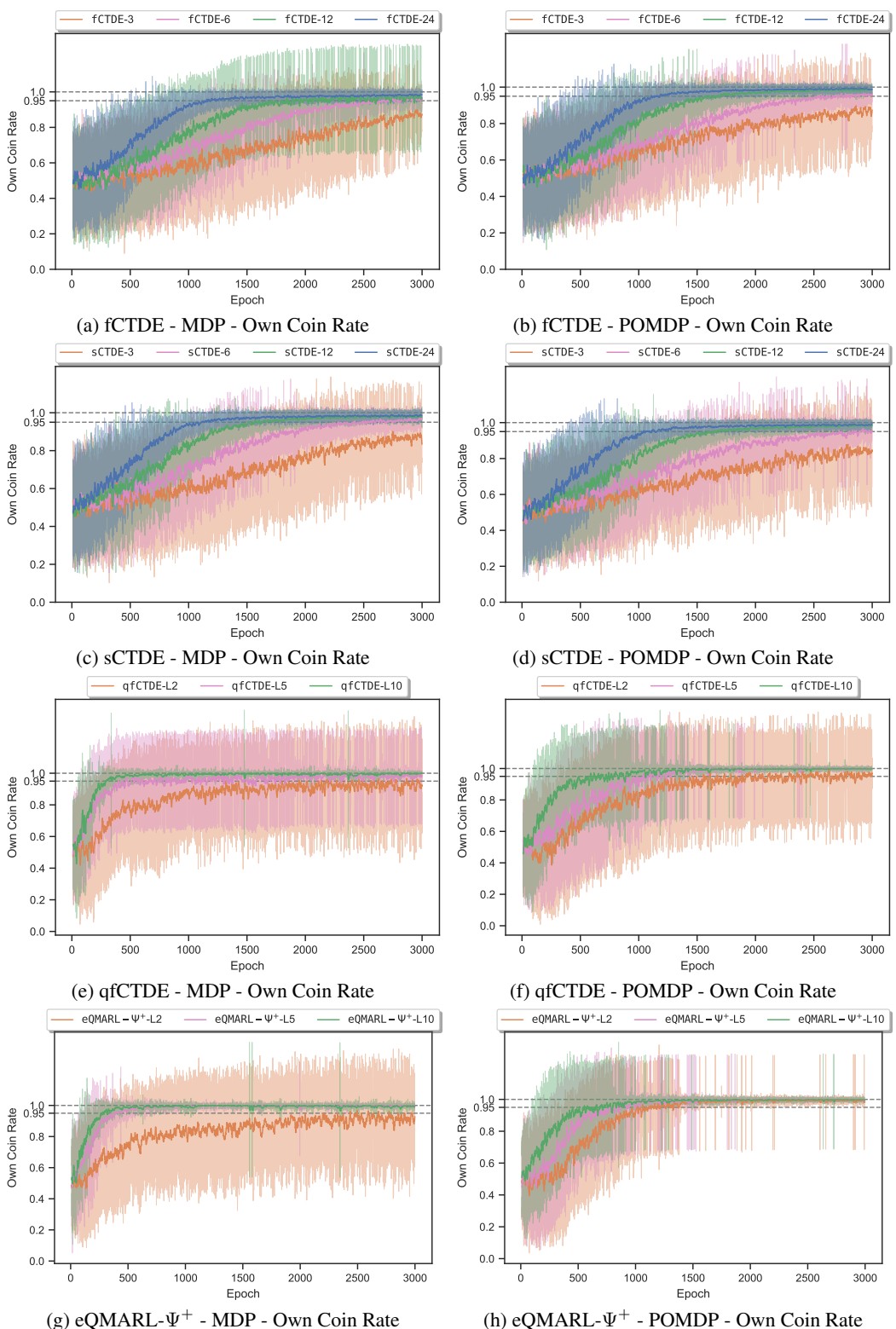

Figure I.6: Own Coin Rate performance for ablation study using `CoinGame-2` for (a,b) fCTDE, and (c,d) sCTDE, and (e,f) qfCTDE, and (g,h) eQMARL-$\Psi^+$ with hidden layer units $h \in \{3, 6, 12, 24\}$ and VQC layers $L \in \{2, 5, 10\}$, averaged over 10 runs of 3000 epochs, with $\pm 1$ std. dev. shown as shaded regions.

