# OpenReview forum: "eQMARL: Entangled Quantum Multi-Agent Reinforcement Learning for Distributed Cooperation over Quantum Channels"
_ICLR.cc/2025/Conference — ICLR 2025 Poster_

### Official Review · Reviewer_xBPK · 2024-10-22

**Soundness:** 3
**Presentation:** 3
**Contribution:** 2
**Rating:** 6
**Confidence:** 2

**Summary:**

Distributed multi-agent reinforcement learning (MARL) faces challenges in balancing the benefits of player coordination with the communication and computational costs of sharing data between players. Quantum mechanics offers potential for more efficient collaboration through quantum entanglement mediated information sharing, but most current MARL approaches still rely on classical information sharing. This paper introduces a novel framework called entangled QMARL (eQMARL), which eliminates local observation sharing by leveraging quantum entanglement, an input entangled state is prepared and joint measurements are performed. A quantum split critic across agents enables joint observation-value estimation without any additional classical communication overhead to share local environment observations. Experimental results show eQMARL converges faster in some cases and with fewer parameters on a central server compared to classical and quantum baselines.

The authors perform experiments on CoinGame and CartPole environments and run these as MDP and POMDP processes using different scoring metrics. Several classical and quantum variants of MARL are benchmarked and compared for converge behaviour, stability, robustness, solution quality and space complexity. Finally, an ablation study on network size supplements the main results and provides insight into the scaling of solution quality with the number of network layers.

**Strengths:**

The paper describes both a novel framework and a novel algorithm which clearly outline how you can train decentralised policies over quantum channels with entangled input qubits. All that is required are joint measurements without sharing local observations amongst agents or a central server. The framework and the technical details are well outlined in Sec 2. with a detailed exposition of the entanglement preparation and the different architectural components which are outlined in Fig. 2. This makes it clear to understand its contribution which lies in the incorporation of quantum channels. As outlined, these do not require explicit agent-server observation reporting and this is a novel Ansatz which has not been explored in the literature.

The benchmarks provided are comprehensive and well presented. Especially in Fig.4 for the CoinGame comparison the eQMARL shows superior convergence dynamics and also improved scores and higher own coin rates. The observation that entanglement stabilises the network is also well put and particularly interesting. Another strong point that is clearly outlined is the idea that additional information in the case of POMPDP would supposedly result in superior performance, but actually it suffices to use entangled input states to generate good learning dynamics. This is interesting for a broader audience.

The work's significance is also clearly outlined. For the benchmarked examples the eQMARL algorithm requires fewer centralised parameters and exhibits more stable and robust learning dynamics than the alternatives. In some cases it also achieves a better solution.

**Weaknesses:**

The results in Fig. 3 are confusing and deserve a clearer exposition. Given the "None" benchmark performs similarly well to all the other Bell states (presumably no entanglement? but this is not clear from the text), how one can understand the role played by the entanglement if it does not significantly outperform "no entanglement"? It would also be interesting to understand why Bell states \Sigma^- and \Sigma^+ perform differently and what role is played by the relative phase?

The cart pole benchmark does not seem to conclusively show an advantage associated with using the eQMARL framework since in both MDP and POMDP cases the authors algorithm does not achieve the maximum reward and also does not seem to clearly outperform qfCDTE (MDP) or qfCDTE and sCDTE (POMDP)even though in the text an advantage - especially in terms of a high mean average reward at early epochs - is claimed to exist. The authors should consider revising the last paragraph and making it less strongly in favour of the performance of the eQMARL algorithm.

There is no mention of the time complexity of the novel framework and particularly the challenge associated with running quantum circuits and preparing entangled input states (notwithstanding the challenge of running this in a non-simulated real quantum device). This makes the benchmarks which all run for the same number or epochs less clear. Particularly in Fig.5b the authors claim the advantage of eQMARL lies in the fact that it achieves a higher reward at a much lower epoch number, but without any information on the per epoch runtime of the different algorithms this case becomes less strong. For example if sCDTE runs 10 times faster than this would not make a strong case for preferring eQMARL over sCTDE. The authors should also consider adressing this in Section 3 when discussing the scaling and performance for different network sizes.

**Questions:**

In the last paragraph the authors claim that "the overall system size is larger, however, the number of central parameters is significantly reduced in the quantum cases – requiring only 1 parameter instead of 25." - can the authors dive into a space time tradeoff here, because no information about algorithm runtimes is given? Even if the eQMARL has fewer centralised parameters, what is its runtime compared to the alternatives? A discussion of this would also be important when comparing the different algorithms in Figs 3-5.

Does the preparation of the input entangled states introduce some computational overhead, can this be quantified? How does this compare to the benchmarked alternative algorithms?

Why does \Sigma^+ perform better than \Sigma^- in the discussion of Fig.3 ? It is only the relative phase which changes not the general entanglement structure.

Why does the None benchmark in Fig.3 perform almost as well as the \Sigma^+ benchmark, is entanglement even required to achieve any advantage and why can you achieve a high score without entanglement?

---

> ### Author Response · Authors · 2024-11-22
>
> # Weaknesses
>
> ## W.1
>
> We have addressed these concerns, which also parallel your questions Q.3 and Q.4, in the global rebuttal section discussing entanglement style performance.
>
> ## W.2
>
> We thank the reviewer for their comments regarding the performance of the multi-agent CartPole experiments. The results from Fig. 5 are indeed not conclusive in isolation, and must be considered along with statistics from Tables H.5 and H.6. From this, there are several key takeaways. In Fig. 5, we see that neither of the classical baselines can achieve a high reward in the MDP scenario, and while sCTDE does achieve the highest score in the POMDP case it takes significantly longer converge than both of the quantum approaches. This shows a clear advantage in the learning efficiency of QMARL as both quantum approaches converge much quicker to a relatively high reward. The specific advantage for eQMARL compared to qFCTDE comes from the similar performance without explicit observation sharing. From Table H.5, we see that eQMARL has a much lower SD than qfCTDE in the MDP case, and while the SD is nearly identical in POMDP we see that eQMARL has both a higher-valued and narrower-ranged CI than qfCTDE. These numbers show that, by using quantum entanglement to remove explicit observation sharing between agents, we can achieve more stable performance in MDP, and similar performance in POMDP compared to qfCTDE. Further, there is an additional benefit in terms of scalability. As shown in Table 1, the composition of qfCTDE requires the entire critic model to reside on the central server -- as the number of agents increases, so does the size of the critic model. In eQMARL, however, the critic model is split across the agents in the system, and only requires a small fixed number of centralized parameters for purposes of quantum measurement. This centralized measurement component does not grow with the number of agents, thereby making it inherently more scalable than all of the baselines. Thus, in summary, the benefits of eQMARL are shown across performance metrics, scalability of model deployment, and the removal of observation sharing. Importantly, these are all new observations that were not studied in prior works.
>
> ## W.3
>
> We thank the reviewer for their concerns on time complexity, and have addressed these points along with question Q.1 below.

---

> ### Author Response · Authors · 2024-11-22
>
> # Questions
>
> ## Q.1
>
> We acknowledge the reviewer's comments regarding time complexity, and agree that stating the training time of eQMARL is valuable information. We will state the training time of eQMARL in the revised manuscript.
>
> The comparison of complexity in terms of "run time" (i.e., wall-clock time) between eQMARL and classical baselines, however, is not a clear apples-to-apples comparison. Indeed, current state-of-the-art QMARL works like Yun et al. (2022a; 2023b; 2022b); Chen (2023); Park et al. (2023); Kölle et al. (2024) do not use wall-clock time complexity as a metric for baseline comparison. Instead, the number of trainable parameters (i.e., model size) is used as the metric for computational complexity, and the training time in units of experiment epochs is used as the metric for relative time complexity. The use of computational and relative epoch time complexity in place of wall-clock time is due to the limitations of running quantum simulations on classical hardware, which is known to be significantly slower in terms of run time than purely classical approaches. In the future, however, this will not be an issue because increased quantum hardware adoption and availability will eliminate the need for quantum simulations on classical hardware; thereby reducing the run time complexity to nearly zero.
>
> Our work is, thus, aligned with Yun et al. (2022a; 2023b; 2022b); Chen (2023); Park et al. (2023); Kölle et al. (2024) and adopts the model size metric to more accurately compare the complexity of our proposed framework with both quantum and classical baselines.
>
> Indeed, the wall-clock training times of both eQMARL and the quantum qFCTDE baseline are significantly slower than those of the classical baselines; which, as previously discussed, is due to the increased overhead of quantum simulations. That said, the training time of both eQMARL and qFCTDE are similar to each other, which, in terms of wall-clock time, is a more apples-to-apples comparison. The results of our ablation study show that the computational complexity of eQMARL is significantly reduced in two key ways. First, the total model size (i.e., agent plus centralized) can be reduced in certain cases, as shown in the MDP scenario, requiring fewer resources to train overall. Second, the number of centralized parameters are both significantly reduced and remain static regardless of the number of agents in the system, as shown in both the MDP and the POMDP scenarios. This is important for scalability because it eases computational burden of the central measurement server compared to the classical sCTDE baseline (which has a centralized component that grows exponentially with agent size).
>
> That said, we can indeed make a clearer note of the "time" complexity of the quantum frameworks and report the run time of our approach in the revised manuscript. As an overview, we conduct all experiments on a high-performance computing cluster with 128 AMD CPU cores and 256 GB of memory per node. The wall-clock training time for classical sCTDE for the CoinGame MDP is $\approx 5.5$ minutes, and for POMDP is $\approx 7.5$ minutes. In contrast, the wall-clock training time of eQMARL for the CoinGame MDP is $\approx 3.5$ hours, and for POMDP is $\approx 8.5$ hours. This training time, however, is in line with many current works on QMARL (such as Yun et al., 2022a; 2023b; 2022b; Chen, 2023; Park et al., 2023; Kölle et al., 2024), and as previously discussed, is indicative of the known computational complexities of running quantum simulations on classical hardware.
>
> ## Q.2
>
> The generation of entangled Bell states is a purely quantum operation using static (i.e., non-parameterized) quantum gates, which are not optimized as a part of the model. As such, the inclusion of entanglement generation does not affect the classical computational complexity of model training, nor does it affect the comparison of computational complexity between the classical and quantum approaches.
>
> ## Q.3 & Q.4
>
> In the reviewer's questions Q.3 and Q.4, the reviewer has used the notation “\\Sigma” ($\Sigma$) which was not used in the paper. Based on the question context we assume you meant “\\Psi” ($\Psi$). We discuss the reasoning for $\Psi^+$ performance in the paragraph starting on line 350 of the manuscript. Further, we have also addressed the analysis of entanglement style performance, and particularly that of $\Psi$ and None, in the global rebuttal. In a nutshell, we believe the performance of None is similar to $\Psi^{+}$ in certain cases due to the degradation of fidelity (i.e., entanglement strength) within the agent variational quantum circuits (VQCs). Further, $\Psi^{+}$ performs better than other entanglement styles due to a combination of opposite-state entanglement ($|01\rangle$ and $|10\rangle$) and no phase shift, which results in less overall fidelity degredation.

---

> > ### Comment · Reviewer_xBPK · 2024-11-25
> > **Response to Author Comments, Time Complexity and Entanglement Styles**
> >
> > Thank you for providing these helpful clarifying comments.
> >
> > Regarding W3 /Q1, I would like to see explicit runtime numbers in the revised manuscript which I have not been able to locate.
> >
> > The point about the different entanglement styles is very interesting and I believe it would be important to highlight this in the main text. The argument which stems from the entanglement degradation of different input states deserves to be fleshed out. Some more theoretical analysis, which concretely explains the phase dependent fidelity degradation for different input states would significantly improve the manuscript and as claimed by the authors would truly shed  ".. new light on the behavior of entangled input states in VQCs, specifically with application to quantum machine learning, that was not known in prior work." I feel as though the experiments performed in Fig. 3 do not conclusively show that, in general, \Psi^+entanglement exhibits superior behaviour to other entanglement styles.

---

> > > ### Author Response · Authors · 2024-11-26
> > >
> > > We thank the reviewer for their comments, and agree on points regarding run time complexity and the discussion of entanglement style performance. As such, we have added remarks and explicit numbers for run time complexity to Section 3.2 (lines 311-316), and a discussion regarding the role entanglement fidelity plays on performance to the last paragraph of Section 3.3 (starting on line 354) in the revised manuscript.
> > >
> > > Your comments regarding more theoretical analysis are well taken. We agree that more theoretical analysis would indeed enhance the manuscript, but having tried to look at this proof since we received the rebuttal, we realized that it is challenging to complete within this rebuttal period as we discuss next. Also, even without this proof, the merits of our approach remain important as will be evident from our discussion.
> > >
> > > As mentioned in our rebuttal, and now in Section 3.3 of the revised manuscript, degradation of entanglement fidelity is a contributing factor to the poor performance of certain entanglement styles. The fidelity between two quantum states $\rho$ and $\sigma$, which are density matrices, is defined as $F(\rho, \sigma) = (\mathrm{Tr}( \sqrt{\sqrt{\rho} \sigma \sqrt{\rho}} ))^2$ from Nielsen & Chuang (2012). The analysis of its impact requires finding an expression for the fidelity between all agents in the system as a function of their final output states as density matrices. In eQMARL, these density matrices represent the final output states of agent VQCs. If, for example, the arbitrary state of an agent is $|\psi\rangle$, then its density matrix is given by the outer product of the state vector with itself, which is $\rho = |\psi\rangle \langle\psi|$, where $\langle\psi| = (|\psi\rangle)^\dagger$ is the complex conjugate. Note that the size of this density matrix is $2^d \times 2^d$, where $d$ is the number of qubits assigned to an agent. These output states are in of themselves functions of the input entanglement unitary $ENT^B_{\delta(1,d),\dots,\delta(N,d)}$ as given by (1), and the split quantum critic, which is a cascade of parallel agent VQC unitaries $U_{vqc}$ as given by (2) in the manuscript. Each agent unitary is both dynamic in layer depth $L$, and parameterized by the agent observations $o^{(n)}$, $\theta^{(n)}$, and $\lambda^{(n)}$, $\forall n \in \mathcal{N}$. Additionally, since the input state is entangled, the resulting output density matrices must be expressed as a function of the Hilbert space of the entire system, i.e., the system density matrix involving the unitaries of all agents simultaneously, which has size $2^{d \times N} \times 2^{d \times N}$, where $N$ is the number of agents, and then reduced to single-agent density matrices using the partial trace operator. This means that, to find an analytical expression for the fidelity between agent states, first an expression for the final $2^{d \times N} \times 2^{d \times N}$ output density matrix as a function of the input entanglement style and $N$ parallel instantiations of $U_{\mathrm{vqc}}(\theta^{(n)}, \lambda^{(n)}, o^{(n)})$ are needed, and then $N$ subsequent partial trace operations are required to obtain expressions for individual agent output density matrices. Simply put, this analytical expression for agent output states is both nontrivial to setup and computationally intractable to solve on classical hardware, even for a small number of qubits and agents. Hence, a complete theoretical analysis regarding the impacts of entanglement fidelity on the coupled system is infeasible given the timeline of this rebuttal period.
> > >
> > > That said, despite the intractability of the theoretical analysis, the merits of our approach can still be seen through the empirical evidence provided in the manuscript, which are important contributions in the development of QMARL. Chiefly, we have shown that maximally entangled Bell states can be used as a coupling medium between agents to improve both their learning efficiency and overall performance while simultaneously reducing computational complexity and eliminating observation sharing overhead. Further, the investigation of agent behavior as a result of different input Bell states across various environmental conditions is also a critical contribution of our work, which was not known before. Future works could expand upon our proposed maximally entangled input state configuration and investigate the theoretical guarantees of both the Bell states and other entanglement styles in environments with an increased number of agents. Moreover, since the choice of entanglement style is seemingly correlated with the nuances of the environment, a logical next step is treat the input entanglement as a learned feature of the joint critic; similar to learned embeddings in a classical setting. Entanglement optimization would in effect tune the fidelity (i.e., strength) of the input state to allow for real-time adjustment of agent coupling based on local environment observations.

---

> ### Comment · Reviewer_xBPK · 2024-11-27
> **Reply to revised manuscript and author comment**
>
> Thank you for this detailed and clear follow up.
>
> I appreciate the effort that went into the detailed discussion of different types of input entanglement and concede that this is not a straightforward question to answer in the scope of the rebuttal period. As stated by the authors, I agree with their claim that "the investigation of agent behaviour as a result of different input Bell states across various environmental conditions is also a critical contribution of our work, which was not known before." and hope that they or others follow up with a more thorough theoretical analysis, verifying or expanding on the here presented empirical results.
>
> I have looked at the revised manuscript and believe that our discussion has improved its quality and clarity. I have updated my score accordingly.

---

### Official Review · Reviewer_H1NG · 2024-11-03

**Soundness:** 3
**Presentation:** 2
**Contribution:** 2
**Rating:** 6
**Confidence:** 3

**Summary:**

This paper proposes eQMARL, a novel quantum multi-agent reinforcement learning framework that uses quantum entanglement to enable distributed cooperation between agents, demonstrating faster convergence (up to 17.8%) and better performance compared to classical and quantum baselines while requiring 25 times fewer centralized parameters.

**Strengths:**

1. This paper seems to be the first to use quantum channels and entanglement for agent coordination, rather than just replacing classical neural nets with quantum circuits. The split quantum critic design that maintains cooperation without observation sharing seems novel to me.

**Weaknesses:**

1. The approach requires sophisticated quantum hardware for generating, maintaining, and transmitting entangled states across agents - capabilities that current NISQ devices don't reliably support.

2. The method's scalability remains unclear, as it's only demonstrated with simple environments and few agents, and could face significant challenges with quantum decoherence and reduced entanglement fidelity as the system grows.

3. The experiments have too small scale for a purely empirical paper.

4. (Minor) you should specify in the main paper that $\otimes$ is the Kronecker (tensor) product.

**Questions:**

1. How does the approach handle agent failures or dropped quantum connections? The current design seems to assume perfect quantum channels and continuous agent availability.

2. How does the choice of entanglement scheme $(\Psi^+, \Psi^-, \Phi^+, \Phi^-)$ affect the information sharing capacity between agents? Why $\Psi^+$ performs best?

**Details Of Ethics Concerns:**

No ethics concerns.

---

> ### Author Response · Authors · 2024-11-22
>
> # Weaknesses
>
> ## W.1
>
> We have addressed these concerns in the global rebuttal document. In short, many recent works discuss entanglement generation and storage for the quantum internet, and most all current works on quantum ML have similar hardware requirements to ours. In other words, eQMARL does not require any special hardware beyond those needed by any existing quantum ML approach in both current and prior research. We hope the discussion in the global rebuttal regarding quantum hardware addresses your concerns.
>
> ## W.2
>
> We address your concerns regarding scalability as part of W.3 below.
>
> ## W.3
>
> The CoinGame and multi-agent CartPole environments were selected to showcase how the presence, or lack, of shared information affects agent learning in scenarios where information sharing is known to be critical or unnecessary to solve the task.
>
> The 2-player CoinGame was previously used in both quantum and classical MARL settings in many seminal works like Foerster et al. (AAMAS, 2018); Phan et al. (AAMAS, 2022); Kölle et al. (2024) as a benchmark for learning cooperative strategies. The 2-player CoinGame is a meaningful baseline for our framework because of its cooperative nature and the ability to restrict information from the players, emulating settings where information sharing between agents is either not possible or desirable (e.g., requiring specialized hardware, network bandwidth limitations, etc.).
>
> The multi-agent CartPole environment is also a meaningful baseline because information sharing across agents in this scenario is not strictly necessary. Agent environments are independent in this scenario, but are governed by the same physical rules. This means that the observations made by one agent (in terms of pole angle, cart direction, etc.) could theoretically be beneficial for the learning of other agents. In the baseline RL settings, these observations are shared amongst agents through explicit communication. In eQMARL, however, we show these same learning benefits can be achieved without explicit observation sharing. Additionally, this application of the multi-agent CartPole environment also shows diversity of experiments for our proposed approach.
>
> That said, to demonstrate an even wider scale of application we have already performed an additional experiment with a multi-agent variant of the MiniGrid environment that showcases the merits of our approach. The results of this additional experiment are given in the global rebuttal. In summary, eQMARL is able to learn a solution with nearly 50\% fewer environment steps, and 4.5-times higher average reward, compared to all baselines.
>
> In terms of scalability with number of agents, we were not able to run the environments with more agents given the time constraints of this rebuttal period, which would require hyperparameter tuning to find optimal parameters. Importantly, however, there are no fundamental technical limitations preventing us from scaling to more agents. In fact, as mentioned in the global rebuttal, eQMARL requires fewer qubits per agent for CoinGame compared to the approach in Kölle et al. (2024), and is aligned with the scalability of previous QMARL works Yun et al. (2022a; 2023b; 2022b); Chen (2023); Park et al. (2023) that require between 4 to 10 qubits per agent.
>
> ## W.4
>
> We agree, and we will clarify the notation in the revised manuscript.

---

> ### Author Response · Authors · 2024-11-22
>
> # Questions
>
> ## Q.1
>
> This is correct, in this work we assume an ideal quantum channel environment and continuous agent availability. These assumptions are identical to those made in the current state of the art QMARL works Yun et al. (2022a; 2023b; 2022b); Chen (2023); Park et al. (2023).
>
> ## Q.2
>
> We discuss the reasoning for $\Psi^+$ performance in the paragraph starting on line 350 of the manuscript. We also provide a further analysis of this in the global rebuttal. In a nutshell, $\Psi^{+}$ performs better due to a combination of opposite-state entanglement ($|01\rangle$ and $|10\rangle$) and no phase shift. The empirical results show that opposite-state entanglement style of $|01\rangle$ and $|10\rangle$ used in both $\Psi^{+}$ and $\Psi^{-}$ perform better (faster convergence and higher score) than same-state entanglement $|00\rangle$ and $|11\rangle$ used in $\Phi^{+}$ and $\Phi^{-}$ in both the MDP and POMDP cases. Further, the phase shift in $\Psi^{-}$ results in a decrease in performance (slightly less overall score and slower convergence) compared to $\Psi^{+}$ in the POMDP case specifically. Therefore, $\Psi^{+}$ with opposite-state entanglement and no phase shift gives the best overall performance in both MDP and POMDP cases compared to the other entanglement styles.
>
> In this work, we selected the maximally entangled Bell states (i.e., $\Phi^{+}$, $\Phi^{-}$, $\Psi^{+}$, $\Psi^{-}$) as a proof of concept due to their relative ease of generation and prevalence in many diverse quantum computing tasks. Our proposed eQMARL framework is not limited to maximally entangled states, however, and the input entanglement gate definition in equation (1) could be modified to generate any other form of entanglement. For example, future works could investigate the use of more complex "weakly" entangled states as input qubits to the agents. These weakly entangled states would still bind the behavior of agent policies, but would interact with the local circular entanglement and variational layers in a different manner.

---

> > ### Comment · Reviewer_H1NG · 2024-12-02
> >
> > Thank you for your response. My main concern is resolved. I decide to raise my score.

---

> ### Comment · Area_Chair_fnUx · 2024-11-27
>
> Dear Reviewer,
>
> The authors have provided their rebuttal to your comments/questions. Given that we are not far from the end of author-reviewer discussions, it will be very helpful if you can take a look at their rebuttal and provide any further comments. Even if you do not have further comments, please also confirm that you have read the rebuttal. Thanks!
>
> Best wishes,
> AC

---

### Official Review · Reviewer_nHF3 · 2024-11-07

**Soundness:** 3
**Presentation:** 2
**Contribution:** 3
**Rating:** 6
**Confidence:** 2

**Summary:**

The paper proposed a novel framework for multiagent RL, based on quantum entanglement. As compared to classical CTDE approaches wherein the communication between the centralized critic and agents happens in the form of sharing observations / states, the proposed framework uses quantum entanglement for information transfer. The framework uses a split critic such that the global value is calculated by transfer of information to a central server via quantum entanglement. The authors present results on two tasks - CoinGame and multi agent cartpole, demonstrating the effectiveness of the proposed framework in sample efficient learning.

**Strengths:**

* The framework proposed is novel and the paper explores the rather unstudied area of using quantum entanglement for information sharing in Multi-Agent RL
* The framework distributes computations across agents, which decreases the centralized computational burden thereby making the framework more scalable
* The proposed framework minimizes the communication overhead between the agents and the centralized critic as well as among the agents via quantum entanglement, as compared to classical information sharing in normal CTDE approaches.
* The empirical evaluation includes a nice study justifying the choice of entanglement type.

**Weaknesses:**

* The paper is not easily understandable to a reader not completely familiar with the area of quantum entanglement. The appendix provides the necessary background on this. However, a brief introduction of the same (a compressed version) in the main paper might provide some context to the reader in order to better understand the paper.
* Some parts of the experimental results section (specifically, lines 363 - 374 and 412 - 425 Section 3.4 and lines 457-464 in Section 3.5) are rather poorly written. Focusing more on the general conclusions as compared to specific numbers and focusing on one metric at a time (either sample efficiency OR absolute performance)  could help improve the readability of the section.
* Although the proposed framework results in lower variance across seeds, the results on multi-agent cartpole task don't seem very encouraging. The proposed framework fails to beat the baselines in the case of both MDP and POMDP
*  The plots in Figure 3 do not immediately make the conclusions clear. Replacing Figure 3 with Tables H.1 and H.2 could be more compelling.

**Questions:**

* From the description provided in Appendix E.2, the multi-agent cartpole does not seem to benefit much from coordination among different agents, since all the agents operate independently. Given that, is it really a good task for evaluating the framework?
* Is it feasible to run experiments with environments consisting of higher dimensional observation spaces. For eg. running experiments on some moderately challenging MiniGrid environments would be interesting.
* Can the authors provide details on the compute required for performing the experiments?

---

> ### Author Response · Authors · 2024-11-22
>
> # Weaknesses
>
> ## W.1
>
> We agree, and we will update the discussion on entanglement in the main paper to include a brief introduction.
>
> ## W.2
>
> We agree, and will revise the experiment results section to favor discussion on insights as apposed to numerical results.
>
> ## W.3
>
> We thank the reviewer for their comments regarding the performance of the multi-agent CartPole experiments. We agree that the efficacy of eQMARL is not readily gleaned from Fig. 5, because the advantage requires an in-depth analysis of both the Figure and the reward metric statistics (which are given in Tables H.5 and H.6). The results for the multi-agent CartPole experiments, however, show that eQMARL learns a similarly performant, and dramatically more stable, strategy than the fully-centralized quantum baseline. This is significant because eQMARL achieves this without centralized access to all agent observations, and with a much smaller centralized critic network size.
>
> ## W.4
>
> Your comments regarding the apparent conclusions drawn from Fig. 3 are well-taken. Similar to your comments regarding W.3 above, the advantages of entanglement style performance requires an analysis of both Fig. 3 and the metric statistics shown in Tables H.1 and H.2. That said, we believe the visual representation of Fig. 3 is more impactful compared to the numerical data shown in Tables H.1 and H.2. In particular, Fig. 3 gives the reader a clearer overview of model performance over time, which is an important distinguishing factor in the analysis of the entanglement styles; a comparison that is less clear from Tables H.1 and H.2 in isolation.
>
> # Questions
>
> ## Q.1
>
> Yes, the multi-agent CartPole environment is a good task for evaluation because the agents can benefit from shared experiences. In multi-agent CartPole the individual agent environments are indeed independent, but follow the same physical rules. This means that the observations made by one agent (in terms of pole angle, cart direction, etc.) could be beneficial for the learning of other agents. In the baseline RL settings, these observations are shared amongst agents through explicit communication. In eQMARL, however, these same learning benefits can be achieved without explicit observation sharing. Further, this difference in shared observation relevance presents a entirely different use case than the CoinGame baseline; hence, using it allows us to show both the application and advantage of our proposed approach in different environments.
>
> ## Q.2
>
> Yes, eQMARL can indeed be run on environments with higher dimensional observation spaces. To demonstrate this, we have already performed an additional experiment with a multi-agent variant of the MiniGrid environment, which showcases the merits of our approach. The results of this experiment are given in the global rebuttal. In summary, eQMARL is able to learn a solution with nearly 50\% fewer environment steps, and 4.5-times higher average reward, compared to all baselines.
>
> ## Q.3
>
> We conduct all experiments on a high-performance computing cluster with 128 AMD CPU cores and 256 GB of memory per node. The wall-clock training time for classical sCTDE for the CoinGame MDP is $\approx 5.5$ minutes, and for POMDP is $\approx 7.5$ minutes. In contrast, the wall-clock training time of eQMARL for the CoinGame MDP is $\approx 3.5$ hours, and for POMDP is $\approx 8.5$ hours. This training time, however, is in line with many current works on QMARL (such as Yun et al., 2022a; 2023b; 2022b; Chen, 2023; Park et al., 2023; Kölle et al., 2024), and is indicative of the known computational complexities of running quantum simulations on classical hardware. As quantum hardware becomes available, this runtime will be significantly reduced.

---

> > ### Comment · Reviewer_nHF3 · 2024-12-01
> >
> > I thank the authors for addressing my concerns.
> > I appreciate the authors furnishing results on a more complex environment.
> > The runtime while being comparable to existing work on QMARL, the runtime is significantly higher than the existing approaches.
> >
> > Considering the above, I continue to remain skeptical of the usefulness (particularly real-world applicability) of the proposed approach in its current form. However, I appreciate the novelty of the direction and the experimental evaluation. This paper may promote more research in this direction which could be useful.
> >
> > I am increasing my score to 6.

---

> > > ### Author Response · Authors · 2024-12-03
> > >
> > > We thank the reviewer for their comments and for increasing their score.
> > >
> > > To clarify the comments of the reviewer regarding runtime complexity, there is an important distinction between the complexity of *purely classical* and *classically-simulated quantum* ML approaches.
> > >
> > > Simply put, quantum simulations on classical hardware are just that -- **simulations** -- which incur run time penalities compared to native deployments. Importantly, this is not a limitation of the quantum algorithm itself, but rather an understood limitation of running quantum algorithms in a non-native fashion, i.e., on classical computers as a consequence of limited access to quantum hardware. Hence, the juxtaposition of wall clock run times for purely-classical and classically-simulated quantum approaches is not truly a fair, i.e., apples-to-apples, comparison. In fact, as mentioned in both the global rebuttal and the revised manuscript, the hardware requirements and wall clock run time of eQMARL is on part with many of the current state-of-the-art QMARL works (Yun et al., 2022a; 2023b; 2022b; Chen, 2023; Park et al., 2023; Kölle et al., 2024). Indeed, this is the case for all current QMARL works that use quantum simulators in lieu of quantum hardware. In truth, and also as mentioned in the global rebuttal, if wall clock run time is viewed as a mitigating factor between purely classical and classically-simulated quantum algorithms, then *all prior quantum machine learning research would be disqualified*.

---

> ### Comment · Area_Chair_fnUx · 2024-11-27
>
> Dear Reviewer,
>
> The authors have provided their rebuttal to your comments/questions. Given that we are not far from the end of author-reviewer discussions, it will be very helpful if you can take a look at their rebuttal and provide any further comments. Even if you do not have further comments, please also confirm that you have read the rebuttal. Thanks!
>
> Best wishes,
> AC

---

### Author Response · Authors · 2024-11-22

# MiniGrid Experiment

A common theme among reviewers was to provide more varied experiments with complex environments. While we did vary our experiments by running in both the CoinGame and multi-agent CartPole environments, each with both full (MDP) and partial (POMDP) information, we have already run an additional experiment with a multi-agent variant of the MiniGrid environment; thus, showcasing that our approach is still effective across more complex environments. In the MiniGrid environment, the task is to find an optimal grid traversal path from a starting position to a goal using the action set {"turn left", "turn right", "move forward"}. We use a reward shaping schedule of $-1$ for every step taken, $-2$ for standing still, $0$ for not reaching the goal, and $100 \times (1 - 0.9 \times (s / m))$ for reaching the goal, where $s$ is the step count, and $m$ is the max steps. In particular, there are four key factors that make this environment even more complex than our previous baselines. First, agents observe the environment using a limited field of view from their current position and rotational direction; hence, the agents must expend actions to both physically move and visually perceive the environment. Second, because the field of view is limited, the goal position is not always in view; meaning that the agent strategies must also learn to search for the goal position in addition to finding an optimal traversal route. Third, the rotational direction of the agent plays a major role in both grid-world visibility and traversal actions; whereby an agent must learn to optimize the total number of rotation actions (i.e., "turn left" and "turn right") for visual exploration and navigation -- e.g., using a single "turn right" action (fewer steps) instead of three "turn left" actions (increased visibility). Fourth, agent actions are not limited when adjacent to grid-world "wall" positions; meaning that if an agent is facing "forward" to a wall the "move forward" action is still valid (even though clearly not optimal). Critically, the tradeoff between increased visibility at the expense of rotation actions poses complex challenges here, and offers a unique opportunity for implicit observation sharing to improve navigation efficiency.

We trained 2 agents over 1000 epochs with a maximum of 50 steps in isolated $5 \times 5$ MiniGrid environments, where, similar to our other baselines, the only coordination allowed is that afforded by the training protocol -- that is, fCTDE and qfCTDE require full observation sharing, and sCTDE and eQMARL use implicit sharing using split critics. The result of this experiment is shown Fig. R.1 in the accompanying PDF of the supplementary material ZIP archive, filename `global_rebuttal_MiniGrid_experiment_figure.pdf`, and in the table below.

Model | Reward Mean | Reward 95% CI | Number of Trainable Critic Parameters
--- | --- | --- | ---
fCTDE | -63.04 | (-65.16, -61.06) | 29,601
qfCTDE | -85.86 | (-87.03, -84.72) | 3,697
sCTDE | -88.02 |  (-88.69, -87.10) | 29,801
eQMARL-$\Psi^+$ | -13.32 | (-14.68, -11.91) | 3,697

From Fig. R.1 we can see that, for the vast majority of training, the fCTDE, qfCTDE, and sCTDE baselines have an average reward that is clustered near -100; meaning that they exhaust many steps by simply spinning in place (since the maximum step size is 50, and $-2$ is the same-position reward). In contrast, we see that the average reward of eQMARL is spread out higher over the training regime. From the table above we specifically see that eQMARL achieves an average overall reward of $-13.32$, which is 4.5-times higher than the other baselines. Indeed, this negative reward means that eQMARL also expends actions turning in place, but the fact the reward is so close to zero implies these events occur at a vastly reduced frequency than the other baselines.

In addition, we also see that both qfCTDE and eQMARL reduce the overall critic size by a factor of 8 compared to the classical baselines; meaning that they are more computationally efficient.

In testing, eQMARL was able to traverse to the goal in as little as 9 steps, whereas fCTDE required 17 steps, and both qfCTDE and sCTDE were unable to find the goal within the 50 step limit. This is a marked 50\% improvement in the exploration and navigation speed of eQMARL over fCTDE, with the bonus of no observation sharing, and an 8-times smaller overall critic size. Hence, we have shown that eQMARL can indeed be applied to more complex environments, such as grid-world navigation with limited visibility, and provide learning benefits over baselines without the need for observation sharing.

---

### Author Response · Authors · 2024-11-22

# Quantum Hardware and Scalability

We acknowledge with the points made by reviewers regarding the limitations of current NISQ hardware, entanglement generation, and storage. Quantum hardware is evolving and many recent works [a-d], particularly those pertaining to the quantum internet, propose methods for generating and storing entangled qubits. Hence, many of those emerging works can be used to support the type of entanglement required in our system. Also note that many current works on QMARL (Yun et al., 2022a; 2023b; 2022b; Chen, 2023; Park et al., 2023; Kölle et al., 2024) have similar hardware requirements to ours.

The empirical evidence indicating the efficacy of our proposed approach on the 2-player CoinGame in fact demonstrates its feasibility for deployment on near-term quantum hardware. Indeed, eQMARL requires only 4 qubits per agent for the CoinGame-2 environment, whereas the approach in Kölle et al. (2024) requires 6 qubits for the same setting. Further, our framework is aligned with previous QMARL works Yun et al. (2022a; 2023b; 2022b); Chen (2023); Park et al. (2023) which use between 4 to 10 qubits per agent. This relative simplicity indicates its feasibility for deployment on emerging quantum hardware, which range between the order of tens to hundreds of qubits depending on the underlying technology. Examples of current quantum computers with these capabilities include the IonQ Forte with 36 trapped-ion qubits released in 2022, Xanadu Borealis with 216 squeezed-state photonic qubits in 2022, QuEra Aquila with 256 neutral-atom qubits in 2022, and IBM Condor with 1,121 superconducting qubits in 2023. Additionally, the results shown for the multi-agent CartPole experiment, also using 4 qubits per agent, further demonstrate the significance of our approach applied to varied environments. Through the results shown in the manuscript, and those of the additional rebuttal, we have demonstrated that our approach is in fact less demanding than others for CoinGame-2, applicable to both fully observed and information-restricted environments, applicable to varied environment learning tasks, and feasible for deployment on the current generation of quantum computers.

We will put a note regarding how entanglement generation and storage methods are open topics of research in the limitations section of the manuscript. Critically, however, if quantum hardware is considered a mitigating factor in the development of new quantum machine learning algorithms, then all research regarding the subject area, including prior works, will no longer be relevant.

[a] Van Meter, et al. 2022. DOI: 10.1109/qce53715.2022.00055.

[b] Pettit, et al. 2023. DOI: 10.1063/5.0156874.

[c] Lei, et al. 2023. DOI: 10.48550/arXiv.2304.09397.

[d] Azuma, et al. 2023. DOI: 10.1103/revmodphys.95.045006.

---

### Author Response · Authors · 2024-11-22

# Entanglement Style Performance

Regarding the points made on clarifying the performance of entanglement styles as shown in Fig. 3, we can see $\Psi^{+}$ performs better by looking at the combined performance across both MDP and POMDP environment settings. In Fig. 3.a we see that no entanglement (i.e., None) and $\Phi^{-}$ take longer to converge between epochs 500 and 1500, and they have a much wider variance between experiments as shown in the low minimum value in the same epoch range. Further, $\Psi^{-}$ also struggles with wider variance in Fig 3.a. In Fig. 3.b we see that both $\Phi^{+}$ and $\Phi^{-}$ take longer to converge, have a lower overall score, and a wider variance. Likewise, $\Psi^{-}$ also takes longer to converge. The only entanglement style that is consistent across both MDP and POMDP cases is $\Psi^{+}$, which has the combination of smallest variance and highest score in both settings.

One explanation for why the performance of None is similar to $\Psi^{+}$ in certain cases could be the degradation of fidelity (i.e., entanglement strength) within the agent variational quantum circuits (VQCs). The circular entanglement unitary within an agent's VQC, as defined in Jerbi et al. (2021), binds the behavior of qubits within an agent by creating a "weakly" entangled state (i.e., low fidelity) from the preceding variational layer. The result of introducing additional input entanglement could, in some cases, lower the fidelity of that entangled state further, resulting in poor model performance. It also could increase the fidelity, however, as similar to the process of entanglement distillation. We believe this decrease in fidelity is the reason why states like $\Phi^{+}$ and $\Phi^{-}$ perform poorly in most cases. Based on this, we also believe $\Psi^{+}$ performs better than $\Psi^{-}$ because of the decoherence associated with the difference in phase of the $|11\rangle$ term. Importantly, this relative phase, or more accurately a polarization-dependent phase shift, changes the structure of the entangled state entirely. Indeed, it differentiates $\Psi^{+}$ and $\Psi^{-}$ as two completely different bases in the combined Hilbert space. This phase is affected by the downstream circular entanglement unitaries within each agent, and thus results in entirely different measurement outcomes.

Importantly, this analysis is another contribution of our work that sheds new light on the behavior of entangled input states in VQCs, specifically with application to quantum machine learning, that was not known in prior work.

---

### Author Response · Authors · 2024-11-26
**Manuscript Revisions**

We thank all the reviewers for their comments. We have just uploaded a revised manuscript, which includes small enhancements as recommended by the reviewers.

The changes are as follows:

1. Added a brief primer on quantum entanglement to Section 2.1 (lines 139-148).
2. Added remarks, and explicit numbers, regarding run time complexity to Section 3.2 (lines 311-316).
3. Revised the experiment results discussion in Sections 3.3, 3.4, and 3.5 to only report percentages as a single metric along with general conclusions.
4. Revised the discussion regarding the role entanglement fidelity plays on performance in the last paragraph of Section 3.3 (starting on line 354).
5. Revised the related works Section 1.1 to discuss general takeaways in favor of a complete review of related works in Appendix A.

---

### Meta-Review · Area_Chair_fnUx · 2024-12-07

**Metareview:**

This paper proposed eQMARL: Entangled Quantum Multi-Agent Reinforcement Learning for Distributed Cooperation over Quantum Channels. During the initial discussion, the reviewers have general concerns of background being too technical, experiments need more clear explanations, and applications in multi-agent RL. The authors carefully addressed these points during rebuttal, both via replies and also via an updated version of the manuscript. In particular, the authors added preliminaries on quantum entanglement, clarified the experimental results, added an experiment on MiniGrid, added discussion on related work, and many others. These changes were well received by reviewers and the scores all became positive after rebuttal.

Therefore, the decision is to accept this paper to ICLR 2025. In any case, the authors should carefully merge all points raised during rebuttal to the final version of the paper.

**Additional Comments On Reviewer Discussion:**

There were detailed discussions during the rebuttal period, during which the authors carefully addressed the comments from reviewers and revised the manuscript. Reviewers nHF3, H1NG, xBPK all increased their scores.

---

### Decision · Program_Chairs · 2025-01-22

Accept (Poster)